

# Event generation with Sherpa 2.2

Enrico Bothmann[1], Gurpreet Singh Chahal[2], Stefan Höche[3], Johannes Krause[4],
Frank Krauss[2], Silvan Kuttimalai[5], Sebastian Liebschner[4], Davide Napoletano[6],
Marek Schönherr[2], Holger Schulz[7], Steffen Schumann[1*] and Frank Siegert[4]

**1** Institut für Theoretische Physik, Georg-August-Universität Göttingen,
D-37077 Göttingen, Germany
**2** Institute for Particle Physics Phenomenology, Durham University, Durham DH1 3LE, UK
**3** Fermi National Accelerator Laboratory, Batavia, IL, 60510-0500, USA
**4** Institut für Kern- und Teilchenphysik, TU Dresden, D-01062 Dresden, Germany
**5** SLAC National Accelerator Laboratory, Menlo Park, CA 94025, USA
**6** IPhT, CEA Saclay, CNRS, Université Paris-Saclay, F-91191 Gif-sur-Yvette cedex, France
**7** Department of Physics, University of Cincinnati, Cincinnati, OH 45219, USA

⋆ steffen.schumann@phys.uni-goettingen.de

## Abstract

SHERPA **is a general-purpose Monte Carlo event generator for the simulation of particle collisions in high-energy collider experiments. We summarise essential features and improvements of the** SHERPA **2.2 release series, which is heavily used for event generation in the analysis and interpretation of LHC Run 1 and Run 2 data. We highlight a decade of developments towards ever higher precision in the simulation of particle-collision events.**

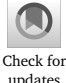
---

# 1  Introduction

Monte Carlo event generators are indispensable tools for the design, realisation, analysis and interpretation of high-energy scattering experiments. In particular, general-purpose generators such as PYTHIA [1], HERWIG [2] and SHERPA [3] are necessary to address detailed aspects of the final states produced in individual scattering events [4]. Typical experimental use cases comprise for example the calibration of object-reconstruction algorithms, the evaluation of detector acceptances, selection efficiencies, or the extrapolation of fiducial cross sections to the full phase space.

Furthermore, over the past decade, Monte Carlo event generators have been established as a tool for precision predictions of scattering cross sections, differential distributions and event topologies. Through the consistent inclusion of higher-order perturbative corrections, in particular in QCD, but also in QED and in the electroweak sector, they nowadays represent state-of-the-art theory calculations that make precision analyses and data interpretation possible. Based on a high level of automation they allow for both the realistic simulation of Standard Model production processes and the description of almost arbitrary New Physics signals. Monte Carlo event generators form a vital cornerstone of collider-based particle physics, from searches for new phenomena to actual Standard Model measurements.

The SHERPA event generator framework, introduced about fifteen years ago [3, 5], is a general-purpose simulation tool for particle collisions at high-energy colliders. It contains implementations of all components needed for a factorised and probabilistic description of scattering events at hadron-hadron, lepton-hadron and lepton-lepton colliders.

This paper summarises the current abilities and components of SHERPA, reflecting the legacy of the SHERPA 2.2 series that was and is being used extensively for the analysis of LHC Run 1 and Run 2 data. A pictorial overview of the SHERPA framework is given in Fig. 1. A generator setup and the corresponding event generation is defined through a text file that contains all non-default settings needed to define the process of interest and to steer the event evolution. The latter includes the setup of the initial beams, the physics model as well as parameters to consider. SHERPA features two built-in tree-level matrix element generators, AMEGIC [6] and COMIX [7,8]. They are used for the simulation of parton-level events within the Standard

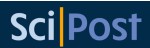

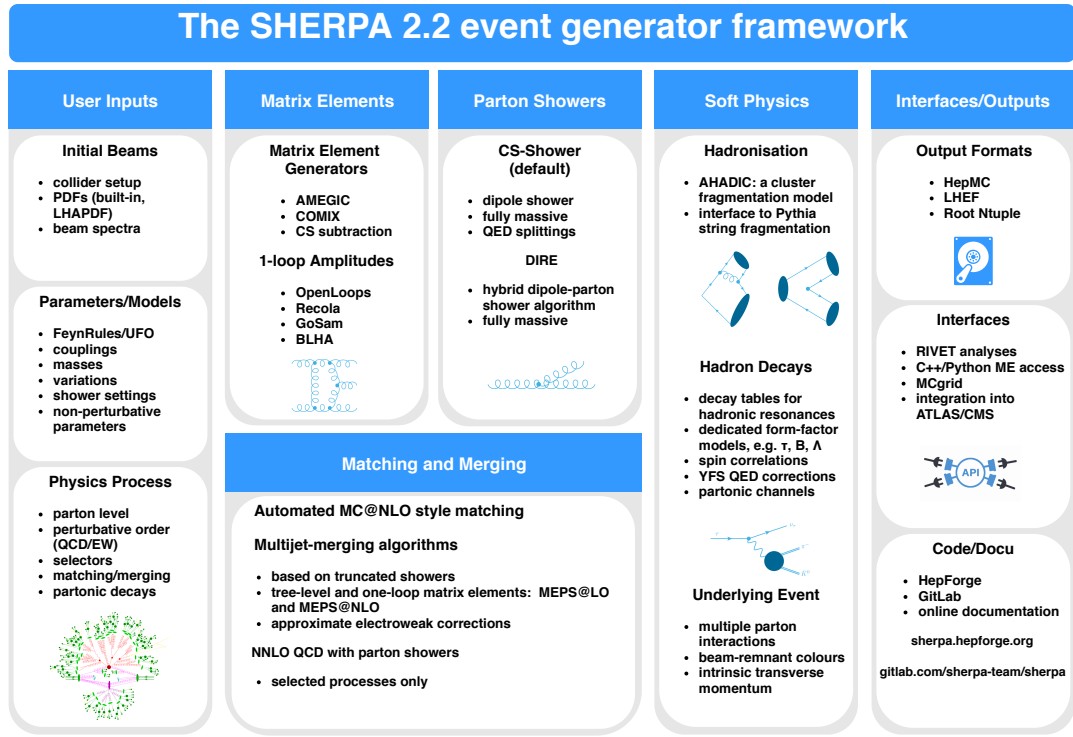

Figure 1: Overview of the SHERPA 2.2 event generator framework.

Model and beyond, and for the decay of heavy resonances such as $W$, $Z$, or Higgs bosons or top quarks. Both include automated methods for efficient phase-space integration and algorithms for the subtraction of infrared divergences in calculations at next-to-leading order (NLO) in QCD [9–11] and the electroweak theory [12]. For the evaluation of virtual corrections at NLO accuracy SHERPA relies on interfaces to dedicated one-loop providers, *e.g.* BLACKHAT [13], OPENLOOPS [14] and RECOLA [15, 16]. The default parton-showering algorithm of the SHERPA 2.2 series is the CSSHOWER [17], based on Catani–Seymour dipole factorisation [9,10,18]. As of version 2.2.0 SHERPA also features an independent second shower implementation, DIRE [19–21]. For the matching of NLO QCD matrix elements with parton showers SHERPA implements the MC@NLO method [22, 23]. For NNLO QCD calculations the UN$^2$LOPs method [24, 25] is used. The merging of multi-jet production processes at leading order [26–28] and next-to-leading order [29,30] is based on truncated parton showers. Multiple parton interactions are implemented via the Sjöstrand–van-Zijl model [31]. The hadronisation of partons into hadrons is modelled by a cluster fragmentation model [32]. Alternatively, in particular for uncertainty estimations, an interface to the Lund fragmentation model [33] of PYTHIA [34] is available. SHERPA provides a large library for the simulation of $\tau$-lepton and hadron decays, including many form-factor models. Furthermore, a module for the simulation of QED final-state radiation in particle decays [35], which is accurate to first order in $\alpha$ for many channels is built-in. To account for spin correlations in production and subsequent decay processes the algorithm described in [36] is implemented. Events generated with SHERPA can be cast into various output formats for further processing, with the HEPMC [37] format being the most commonly used. In the specific case of parton-level events, at the leading and next-to-leading order in QCD, additional output formats are supported. They include Les Houches Event Files [38], NTUPLE files for NLO QCD events [39] and cross-section interpolation grids produced via MCGRID [40,41] in the APPLGRID [42] and FASTNLO [43,44] formats. To analyse

events on-the-fly a runtime interface to the RIVET package [45] can be used conveniently.

The SHERPA Monte Carlo is publicly available from its HEPFORGE project page sherpa.hepforge.org. The actual code development and bug-tracking facilities are hosted on gitlab.com/sherpa-team/sherpa. The current release version is SHERPA 2.2.8.

The paper is organised as follows. Sec. 2 will focus mainly on highlighting and summarising the specific physics implementations and realisations in SHERPA, referring to more in-depth original literature where appropriate. This section also includes a brief discussion on aspects related to the tuning of non-perturbative model parameters in SHERPA. In Sec. 3 we present selected results obtained with recent versions of SHERPA that shall illustrate typical use cases and highlight specific aspects of the simulation. We present our conclusions and an outlook in Sec. 4.

Please note, for more detailed and pedagogical reviews of general Monte Carlo event generation techniques and their practical implementations we refer interested readers to [4, 46, 47].

## 2 Highlighting SHERPA Components

In the following we will briefly describe the central components of the SHERPA framework. We focus on the physics models and features available, providing references to the original literature for more detailed theoretical derivations and discussions.

The SHERPA framework is written in C++in a highly modular structure, reflecting the factorised ansatz to calculate the evolution of scattering events. The SHERPA core module is responsible for steering the event generation process. It initialises the required physics modules and iterates the steps of the simulation. The setup of each generator run, including the specification of model parameters and all switches, is read from a simple ASCII file, called `Run.dat` per default. Parameters of a specific simulation aspect are collated in blocks following a simple bracket syntax:

```
(block_name){
Parameter1 Value1;
Parameter2 Value2;
...
}(block_name)
```

Examples of blocks are (`run`), where general settings are kept, while the specification of the hard scattering process to be considered is compiled in (`processes`). Settings related to cuts on the hard scattering final state are given in (`selectors`). Specific run parameters will be highlighted along with the presentation of the physics models in this section and the examples in Sec. 3. We organise the discussion beginning with methods for the hard-process generation in Sec. 2.1, followed by parton showers and the methods for matching and merging them with higher-order matrix elements in Secs. 2.2 and 2.3, respectively. We present the evaluation of perturbative uncertainties based on a reweighting method in Sec. 2.4. This is followed by a brief discussion of available beam spectra and distribution functions in Sec. 2.5. Section 2.6 is devoted to the discussion of higher-order QED and electroweak corrections in the decays of unstable particles. Sec. 2.7 presents our treatment of beam remnants and the underlying event, while Sec. 2.8 describes the SHERPA cluster hadronisation model. We close by presenting our methods for $\tau$-lepton and hadron decays in Sec. 2.9.

### 2.1 Hard-Scattering Matrix Elements

The simulation of individual events starts from a partonic hard-scattering configuration, with momenta distributed according to the corresponding squared QFT transition matrix element.

Sampling those partonic events allows one to determine the total production rate and differential distributions of the final-state objects to a given perturbative fixed-order accuracy, *e.g.* at tree-level or at next-to-leading order in the strong or electroweak coupling. Given the plethora of processes that users might want to study – both within the Standard Model and various theories for New Physics – a high level of automation is mandatory for the construction and evaluation of matrix elements.

In SHERPA a large variety of fixed-order calculations are available, ranging from the explicit implementation of some simple $2 \to 2$ squared amplitudes at leading order (LO) and next-to leading order (NLO), over automated matrix-element generators (MEGs) for tree-level processes with large multiplicities of external particles, to interfaces to external matrix-element implementations at tree- and one-loop level. The respective MEG to be used in a simulation run is specified via:

```
(run){
...
ME_SIGNAL_GENERATOR Internal Amegic Comix BlackHat OpenLoops Recola ...;
...
}(run)

(processes){
...
Loop_Generator Internal BlackHat OpenLoops Recola ...;
RS_ME_Generator Amegic Comix;
...
}(processes)
```

`ME_SIGNAL_GENERATOR` defines the global choice for the matrix-element provider(s) to be used throughout the run. When specifying several values they are consecutively asked to provide the requested matrix element. By specifying `Loop_Generator` and/or `RS_ME_Generator` the generators for the loop amplitudes and the subtracted real-emission terms may be chosen separately.

**Built-in Matrix Element Generators** SHERPA includes two fully automated MEGs, AMEGIC [6] and COMIX [7], for the calculation of fixed-order total and differential cross sections and decay widths for multi-particle production and decay processes at tree level. Both MEGs are capable of simulating complicated final states as chains of subsequent decays in the narrow-width approximation, including a proper treatment of all effects due to spin and colour correlations. COMIX allows for external particles with spin-0, 1/2, and 1, while AMEGIC also supports external spin-2 particles [48]. In both MEGs Majorana fermions are treated using the formalism presented in [49]. Squared amplitudes in both AMEGIC and COMIX can be projected on arbitrary orders in the contributing couplings. This permits, among others, the computation of pure QCD contributions to the cross section or to exclusively select interference terms, see *e.g.* [51].

To give an example, the definition of the tree-level partonic processes for hadronic electron-positron-pair production in association with two final-state partons reads:

```
(processes){
% use light-jet container 93
Process 93 93 -> 11 -11 93 93;
% constrain orders in strong (1st) and ew (2nd) coupling
Order (2,2);
End process;
}(processes)
```

Note, particles are referred to using their PDG Monte Carlo number [52]. In addition, SHERPA permits the utilisation of both predefined and user-specific particle containers. In the above example, the predefined container 93 comprises all massless QCD partons, *i.e.* gluons and massless quarks. The coupling orders are counted at the squared matrix element level.

The factorisation and renormalisation scales used in the evaluation of the hard process can be specified through

```
(run){
...
SCALES <scale-setter>{<fac-scale-definition>}{<ren-scale-definition>};
...
}(run)
```

Possible scale setters for fixed-order calculations include `VAR` and `FASTJET`. The first allows the use of simple user-defined functions of the final-state momenta, the latter invokes jet finding via FASTJET [53]. In both cases particle/jet momenta are accessible through `p[<i>]`, where `i=0,1` labels the initial-state momenta and final-state particles or $p_T$-ordered jets use `i>1`. Examples to set both the factorisation and the renormalisation scales to either the invariant mass of the two jets or their scalar sum read:

```
% VAR scale setter
SCALES VAR{Abs2(p[4]+p[5])}{Abs2(p[4]+p[5])};

% FASTJET scale setter
SCALES FASTJET[A:antikt,PT:30.,R:0.4,M:0]{H_T2}{H_T2};
```

To regularise the phase space, cuts on the final-state leptons and partons need to be applied. A possible event selection may read (again using the FASTJET package for jet finding):

```
(selector){
% window cut on di-lepton invariant mass
Mass 11 -11 80. 100.;
% transverse momentum cut pT>15 GeV on leptons
PT  11 15. E_CMS;
PT -11 15. E_CMS;
% require at least 2 anti-kt jets with R=0.4 and pT>30 GeV
FastjetFinder antikt 2 30. 0 0.4;
}(selector)
```

AMEGIC effectively performs a colour decomposition of the full amplitude, leading to gauge-invariant subsets of amplitudes for each colour structure. These terms are each composed of Feynman diagrams expressed as helicity building blocks based on [54]. In this amplitude construction process, common sub-amplitudes are identified and algebraically factored out [56], thereby dramatically reducing evaluation times later on. The resulting expressions are written out as C++source code, compiled and linked into dynamic libraries. During the event-generation phase, these libraries are automatically located and loaded to the main code.

COMIX implements the colour-dressed Berends–Giele recursive relations [57], a tree-level equivalent of the Dyson–Schwinger equations [58], to construct off-shell currents that are fused into amplitudes. Information about the valid current and vertex assignments in the process is written to disk in the form of text files such that subsequent runs of the generator requesting the same process can commence faster. COMIX uses the colour-flow representation [61] and colour sampling to compute cross sections including QCD particles. This explicit computation of colour-ordered amplitudes turns out to be advantageous in the context of matrix-element parton-shower matching and merging.

**BSM Simulations**   An interface to FEYNRULES [63–65], *i.e.* the UFO model definition files [66], allows the user to consider a wide range of models. In AMEGIC, however, only vertices with up to four external particles are supported, imposing some limits on its abilities, while in COMIX this number is limited only by computing power, allowing calculations in more complicated theories. With the physics-model information encoded in the standard UFO format [66], SHERPA creates and links complete C++source code necessary to compute arbitrary scattering processes, employing an automatic generator for Lorentz [8] and colour [67] structures which represent the elementary vertices of the theory. The generators AMEGIC and COMIX have been extensively benchmarked, internally and against other codes, for example in the SM [56] and the MSSM [68][1].

**Phase-Space Integration**   SHERPA uses various methods to efficiently integrate multi-particle phase spaces, implemented in its PHASIC module. These can be classified as importance-sampling techniques, where phase-space points are generated using suitable approximations for the desired target distribution, that is given by the squared matrix element. For this purpose a set of phase-space maps (called channels) is automatically constructed by the MEGs according to the propagator and vertex structures of contributing Feynman diagrams or current topologies. The full set of contributing integration channels is combined into a multi-channel integrator that features an automatic optimisation of the individual channel weights [69].

In AMEGIC this leads to the construction of typically one channel per diagram [6]. Within COMIX, the method is recast into a recursive algorithm, reducing the factorial growth in the number of channels to an exponential one [7]. Both AMEGIC and COMIX further optimise the integration over propagator masses and polar angles in decays, using a re-mapping of random numbers base on VEGAS [70, 71] for each channel.

**Resonance Decays**   Intermediate unstable resonances, as they frequently appear in extensions of the Standard Model, can produce high-multiplicity final states through cascade decays. In SHERPA there are two ways of treating such effects. The first is to select solely *s*-channel diagrams/current topologies of the requested intermediate resonances, thereby automatically taking into account finite-width and spin-correlation effects while possibly violating gauge invariance of the overall amplitude. An example for the production and decay of top quarks in electron-positron annihilation reads

```
(processes){
% enforce intermediate top-quarks
Process 11 -11 -> 6[a] -6[b];
% decays t -> bW
Decay 6[a] -> 5 24[c];
Decay -6[b] -> -5 -24[d];
% decays W+ -> mu+ nu, W- -> qqb'
Decay 24[c] -> -13 14;
Decay -24[d] -> 94 94;
End process;
}(processes)
```

Alternatively, employing a strict narrow–width-type factorisation of production and decay, resonances can be produced as external particles and then decayed through separate decay matrix elements. By default, a posteriori the decay kinematics is adjusted to a Breit–Wigner distribution using the resonance's width. Spin correlations are retained through the algorithm worked out in [36, 72–74]. For the latter case, SHERPA automatically constructs the decay tables

---

[1] We would like to note that since version SHERPA-2.0 the realisation [48] of the ADD model of large extra dimensions is no longer supported.

and computes the partial widths and branching ratios at tree level. It is possible for users to overwrite any of the automatically generated branching ratios, and to enable or disable any subset of decay channels. This can be useful, for example, to include NLO K-factors or to better match known (and measured) branching ratios. The setup corresponding to top-quark production and decay from above in the factorised approach reads:

```
(run){
% enable hard decays
HARD_DECAYS On;
% enforce decay W+ -> mu+ nu
HDH_STATUS[24,-13,14] 2;
% switch off decays W- -> l- nu
HDH_STATUS[-24,-12,11] 0;
HDH_STATUS[-24,-14,13] 0;
}(run)

(processes){
% produce final-state tops, decay through hard decay module
Process 11 -11 -> 6 -6;
End process;
}(processes)
```

**NLO Calculations and One-Loop Providers** The inclusion of NLO QCD corrections to a given scattering process has become a de-facto standard in today's event generators, including their matching to parton showers. As both virtual and real-emission corrections are separately infrared divergent, a cancellation procedure is required. In SHERPA this has been realised for the first time through the automation of the Catani–Seymour dipole subtraction formalism [9, 10] in [11]. Renormalised QCD virtual corrections are obtained either through dedicated interfaces from programs and libraries like BLACKHAT [13, 75–77], MADGRAPH [78, 79], OPENLOOPS [14] and RECOLA [80], or through the generic Binoth Les-Houches Accord interface [81, 82] from codes like GOSAM [83, 84] or NJET [85]. An example process declaration including the evaluation of NLO QCD corrections in a fixed-order computation reads:

```
(processes){
Process 93 93 -> 11 -11 93 93;
% use asterisk wild card for strong coupling
Order (*,2);
% evaluate NLO QCD corrections in fixed order scheme
NLO_QCD_Mode Fixed_Order;
% include Born (B), Virtual (V), Integrated Subtraction (I), Real (R) and
    Subtraction terms (S)
NLO_QCD_Part BVIRS;
Loop_Generator <One-Loop Provider>;
End process;
}(processes)
```

Examples of NLO QCD calculations performed with SHERPA include:

- vector-boson production with up to five jets at NLO QCD [86, 87],

- Higgs-boson production in association with up to three jets, taking into account finite-mass corrections [88],

- top-quark pair production with up to three jets [89],

- diphoton plus up to three jets production [90,91],

- up to five-jet production at the LHC [92–94].

The generalisation of the subtraction formalism to electroweak corrections has been implemented in [12] and renormalised electroweak one-loop corrections can at present be obtained from GOSAM, OPENLOOPS and RECOLA. They are, however, not yet available in SHERPA-2.2. Processes that have already been evaluated at full EW one-loop order with a development version of SHERPA include:

- three-jet production at the LHC [95],

- four-lepton production [16,96,97],

- $t\bar{t}h$ production [16],

- $W$ ($Z$) production with up to three (two) jets [16,98],

- $\gamma\gamma W$ and $\gamma\gamma Z$ production [99],

- and $\gamma\gamma j$ production [100].

**NNLO QCD Calculations**    For a few phenomenologically highly relevant processes SHERPA allows for the evaluation at NNLO QCD precision, using the $q_{\mathrm{T}}$-slicing method based on the ideas of [101, 102]. QCD NNLO cross sections can be computed in SHERPA for neutral and charged current Drell–Yan processes [25, 103] and for Higgs-boson production [24].[2] Note, the NNLO facilities are not distributed with the public code releases, but can be obtained in the form of plugins from http://www.slac.stanford.edu/~shoeche/pub/nnlo/.

**External Matrix Elements**    SHERPA provides a generic interface to external amplitude generators, that can be used in particular to compute cross sections for loop-induced processes, like *e.g.* $gg \rightarrow W^+W^-$ or $gg \rightarrow HH$. For matrix elements provided by OPENLOOPS [14], the interface is fully automated and can be used as a blueprint to access other external MEGs as well.
For the phase-space integration of externally provided matrix elements, a set of process-specific phase-space generators is available in SHERPA. If they need to be extended, a phase-space generator for a process with similar characteristics can be generated with AMEGIC and then used as a plug-in. Alternatively, phase space can be sampled uniformly, using the SHERPA implementation of the RAMBO algorithm [105].

## 2.2 Parton Showers

QCD parton showers form an indispensable part of any multi-purpose event generator. They account for the successive emission of QCD or QED quanta off the initial- and final-state partons of the hard process. In doing so, showers relate a few-parton hard-scattering configuration at momentum scale $Q^2_{\mathrm{hard}}$ to a set of partons with typical inter-parton separation scales down to $Q_0^2 \approx 1\,\mathrm{GeV}^2$. This solves the evolution of arbitrary hard-scattering processes from high to low scales, where ultimately a non-perturbative hadronisation process sets in, transforming the final-state partons into primary hadrons.

Formally, parton showers provide approximate numerical solutions for the all-orders resummation of large kinematical logarithms. A statement on the logarithmic accuracy for an

---

[2] A development version of SHERPA has also been used to compute NNLO cross sections for di-boson production at hadron colliders [104].

arbitrary observable evaluated with a shower algorithm cannot easily be made. However, in recent years investigations on the correspondence of parton showers to resummation approaches have been fruitful, see for instance [106–108].

Furthermore, the need to match parton showers to higher-order matrix elements, in particular multi-leg tree-level or one-loop matrix elements, has served as a development paradigm. This raises issues about matching the exact singularity and colour structure of QCD matrix elements, preserving their fixed-order accuracy, without compromising on the resummation property of the parton shower. This has for instance led to the formulation of shower algorithms based on NLO QCD infrared subtraction schemes.

SHERPA comprises two different parton showers, based on different construction paradigms, and implementing different ways to fill the phase space for multiple emissions of secondary particles.

**CSSHOWER**    The default shower of the SHERPA-2 series is based on Catani–Seymour dipole factorisation [9, 10], first proposed in [18]. The technique was implemented in SHERPA [17] and at the same time in [109], building on a set of generic operators for particle emission off a dipole in unintegrated and spin-averaged form in the large-$N_c$ limit. Each dipole contains a splitting parton and a colour-connected spectator parton. The shower evolves through sequential splittings of such dipoles. In the SHERPA implementation, all QCD splittings within the Standard Model and the MSSM, as well as the emission and splittings of photons are incorporated, evolving QCD and QED quanta on an equal footing [110]. Note that the CSSHOWER fully supports finite-mass effects. This is important in particular for the production and evolution of $b$-quarks [17,110], thus allowing for systematic studies of $b$-quark associated/initiated processes in the four- and the five-flavour scheme [111, 112]. With this, the CSSHOWER, in essence, implements a fully-differential general-mass variable flavour-number scheme (GM-VFNS), generating massive quark thresholds through momentum conservation. Furthermore, general electroweak splittings are implemented in CSSHOWER [113]. However, as the chirality of fermions can currently only be treated in an approximated form, these splittings are disabled by default.

In the dipole picture of the CSSHOWER, soft-gluon emissions are mapped onto two dipoles, which consist of the same partons, but with the roles of spectator and emitter interchanged. The splittings are ordered by their associated transverse momenta. For final-state splitters, this is the transverse momentum between the two daughters, whereas for initial-state splitters the transverse momentum is taken with respect to the emitting beam particle. In contrast to the original formulation [17], where the kinematics of the Catani–Seymour formalism is used, in the current default configuration recoil from the emission is either compensated by the spectator if the emitter is a final-state parton, or otherwise distributed equally among all final-state particles. This modified recoil scheme was first proposed in [114] and refined to include massive partons in [110]. It is crucial to obtain reliable predictions for Deep Inelastic Scattering (DIS) processes [110,115].

The above choices were made with the matching and merging of the shower with hard matrix elements in mind (these techniques are described in Sec. 2.3). Building the splitting kernels on top of the subtraction formalism used to calculate NLO matrix elements allows one to write the MC@NLO formalism in the most simple form. Using the transverse momentum as the ordering variable removes the need to veto splittings with scales that are larger than the scale set by the hard process. And finally, local energy-momentum conservation enables the translation of a multi-leg matrix element into a history of parton-shower emissions, which is needed for attaching showers to multi-parton amplitudes [28].

**DIRE**  The second parton shower implemented in SHERPA is DIRE [19], it presents a hybrid between the colour-dipole picture [116] and standard collinear parton evolution. Similar to the CSSHOWER, it is based on Catani–Seymour dipole subtraction [9, 10], but uses the inverse of the soft eikonal as evolution variable. The soft-enhanced part of the splitting functions is defined by a partial fraction of the soft eikonal of the colour dipole [9], giving the correct soft-anomalous dimension at one-loop order. The collinear remainder of the splitting kernels is determined by the constraint that they reproduce the known collinear anomalous dimensions, while respecting flavour and momentum sum rules.

The resulting splitting functions can be negative, leading to negative emission probabilities which necessitate the weighted Sudakov veto algorithm, introduced in [110, 117, 118]. The negative prefactor is then moved to an analytic event weight. In the same way, DIRE can also deal with negative values of PDFs without resorting to an unphysical emission cut-off. The event-weight variance imposed by this approach is typically small.

DIRE uses the same recoil strategy as CSSHOWER, and as for CSSHOWER, massive partons are supported, with the additional construction principle that the evolution variable of DIRE is still mapped to the soft-enhanced term of the full matrix element. A unique feature of DIRE is that it has also been implemented in PYTHIA [1], allowing extensive cross validation between the two generators thus enabling stringent consistency checks of event samples produced for experimental analyses.

Within the framework of DIRE, it has been shown that triple-collinear and double-soft NLO corrections to the splitting functions can consistently be included in a parton shower [20, 119]. A complete treatment of higher-order corrections will be available in a future version of SHERPA.

## 2.3   Matching and Merging

Having discussed the methods used for calculating hard-scattering matrix elements and parton showering raises the question how to combine these two complementary approaches, while preserving their respective strengths. Consider a well-defined inclusive $n$-jet type observable. A tree-level calculation at $\mathcal{O}(\alpha_s^n)$ will typically provide the lowest-order prediction. Subsequent emissions from a parton shower provide a (leading) logarithmic approximation for the higher jet rates, preserving the leading-order $n$-jet rate. In contrast, an exact NLO QCD calculation, *i.e.* including the virtual and real corrections, yields an NLO accurate prediction for the $n$-jet cross section, while the $(n + 1)$-jet rate is approximated to leading order.

*Matching* matrix elements and parton showers resolves the double-counting of the NLO corrections in the matrix-element calculation with the first parton-shower emission. Multijet *merging*, on the other hand, allows for the combination of final states of increasing matrix-element parton multiplicity, evolved by a parton shower, into an inclusive description. This enables prediction for higher jet rates at NLO or LO accuracy, depending on the order of the underlying matrix-element calculation, up to a certain maximum matrix-element parton multiplicity. Yet higher jet numbers are accounted for by the parton shower off the highest-multiplicity matrix element.

Multijet merging, first introduced in [26, 27], has been one of the cornerstones of SHERPA since its inception. Promoting the idea underlying multijet merging to the inclusion of higher-order matrix elements builds on the exact matching of these matrix elements to the subsequent parton showering, delivering precise simulations in their own right. In this section we describe the methods for matching and merging in SHERPA, including the incorporation of NNLO QCD corrections for a few processes and means to account for approximate NLO electroweak contributions relevant in particular in high–momentum-transfer regions.

**Matching of NLO Matrix Elements and Parton Showers**  For the matching of next-to-leading order matrix elements, SHERPA uses a variant of the MC@NLO method [22]. Its basic idea

is the realisation that parton showers organise their radiation pattern, and in particular the first emission, by identifying and factorising the singular soft and collinear limits of the emission matrix elements. In parton showers the notion of a resolution parameter in the emission phase space of the secondary quanta regularises the singularities, leading to the appearance of logarithms in the cut-off parameter. In NLO calculations, however, these singular terms must be identified and subtracted from the real-emission matrix elements. This enables the decomposition of the parton-level calculation into two parts with well-defined, finite cross sections: an infrared-subtracted real-emission contribution, where the subtraction is identified as the first parton-shower emission off an underlying Born configuration, and a part consisting of the original Born-level calculation supplemented with the virtual correction and the integrated infrared subtraction terms, both of which share the same Born kinematics. Parton showers are attached to both parts, with starting conditions reflecting the respective kinematics. In SHERPA, this idea has been recast in a form that maximises the benefit of using identical kernels for infrared subtraction and parton showering [23].

In the past decade, the MC@NLO matching in SHERPA has been continuously developed and refined. Specific aspects and applications have been discussed in a series of dedicated publications, including pure jet production at the LHC [120], the hadronic production of electroweak gauge bosons and up to three jets [121], $t\bar{t}b\bar{b}$ production [122], $s$- and $t$-channel single-top production [123] or Higgs-boson pair production [124]. The MC@NLO approach is nowadays routinely used in Standard Model simulations with SHERPA. Furthermore, it forms the basis for all *merging* approaches involving NLO QCD matrix elements. To give an example, the process definition of an MC@NLO matched simulation of Drell–Yan lepton-pair production in association with two jets reads:

```
(run){
...
SHOWER_GENERATOR CSS
...
}(run)
(processes){
Process 93 93 -> 11 -11 93 93;
Order (*,2);
% evaluate NLO QCD corrections in MC@NLO scheme
NLO_QCD_Mode MC@NLO;
Loop_Generator Recola;
End process;
}(processes)
```

Note, selector definitions similar to the ones stated in Sec. 2.1 apply here as well.

**NNLO Matrix Elements and Parton Showers: First Steps**   Using the UN$^2$LOPs method proposed in [24], and relying on $q_T$-slicing [101, 102] to regulate the additional infrared singularities, it is possible to also include NNLO-correct matrix elements for the production of colour-singlets at hadron colliders into a parton shower framework. In SHERPA this has been achieved for two processes, Drell–Yan and Higgs-boson production [24, 25], thereby providing an important alternative to the MINLO-based implementations of [125, 126]. More recently, the application to hadronic final state production in Deep Inelastic Scattering [127] has been discussed. However, here the projection-to-Born method [128], rather than the $q_T$-slicing technique has been used to regulate the additional infrared singularities appearing at NNLO. Note, the NNLO+PS facilities are not distributed with the public code releases. The Drell–Yan generator can be obtained from http://www.slac.stanford.edu/~shoeche/pub/nnlo/.

**Multijet Merging at LO and NLO**   The multijet-merging approach uses the notion of jets – usually defined through a $k_T$-type measure – to classify emissions as either jet production or jet evolution, and to additively combine towers of exact matrix elements with increasing jet multiplicities into one inclusive sample. Denoting the separation scale of both regimes as $Q_{\text{cut}}$, emissions with $Q \geq Q_{\text{cut}}$ get accounted for by exact matrix elements, while radiation with $Q < Q_{\text{cut}}$ is described by the parton shower instead. In turn, hard jet-emission configurations will follow the fixed-order matrix-element kinematics, while the inner-jet evolution and the production of additional softer jets is in the realm of the parton shower's emission kernels. The resummation of emission-scale hierarchies is provided by the parton shower in both regimes.

The classification into two disjoint, complementary regimes avoids the explicit double-counting of emissions, while the logarithmic accuracy of the parton shower is recovered by both the matrix elements and the parton shower's emission kernels having the same infrared limits (at leading $N_c$) and using the parton shower's resummation in both regions. Originally these ideas have been proposed for the combination of tree-level matrix elements in [26, 27], and have been implemented, in variations, in all event generators [28, 29, 129–136]. A dedicated comparison can, for example, be found in [131].

The SHERPA merging algorithm for tree-level matrix elements, called MEPS@LO, has been detailed in [28]. It relies on a truncated parton shower, *i.e.* the shower explicitly generating the Sudakov form factor for lines between reconstructed matrix–element-type emissions. Broadly speaking the algorithm proceeds as follows:

- initial cross sections for the multijet matrix elements to be considered are evaluated,

- according to the total cross section a specific jet multiplicity is picked, then a flavour channel and an event kinematics are generated,

- for the given flavour assignment and kinematics a clustering algorithm is applied that inverts the parton shower algorithm until a unique core process and subsequent emission scales in the full matrix-element configuration are identified,

- a scale choice is made for the strong-coupling factors, comprising the respective contributions for both the identified core process and the individual emission scales, identical to those used in the parton shower,

- the truncated parton shower starts from the core configuration, reconstructing the identified matrix-element emissions when the shower evolution parameter crosses their predetermined scales, and the event is vetoed when the parton shower produces an emission above the resolution scale $Q_{\text{cut}}$, implementing the Sudakov factor of the parton-shower resummation in the matrix-element region.

This procedure allows one to add event configurations exclusive in the emission scale down to $Q_{\text{cut}}$ into an inclusive sample, thereby cancelling the dependence on the separation parameter to the logarithmic accuracy of the parton shower. Note, the sample of highest matrix-element multiplicity has to be exclusive down to the lowest matrix-element emission scale only, *i.e.* $Q_{\text{last}}^{\text{ME}} \geq Q_{\text{cut}}$.

The well-established LO approach has been promoted to include matrix elements at NLO accuracy in QCD, called MEPS@NLO, and implemented in SHERPA in [30, 137, 138]. It combines MC@NLO matched samples of increasing jet multiplicities, separated by the resolution parameter $Q_{\text{cut}}$ into an inclusive sample. In general, the approach follows the outline above, only the usual care with overlapping descriptions through NLO matrix elements and parton showers is taken, and any overlaps are carefully removed to fully maintain the respective accuracies throughout. Other formulations and approximations thereof have been presented in [139–142].

**NLO EW Corrections in Matching and Merging**   In [143, 144], an approach to incorporate approximate electroweak and subleading mixed QCD-EW corrections into the above described MEPS@NLO QCD method was introduced, dubbed MEPS@NLO QCD+EW$_{\text{approx}}$. There, the Born-like input cross section into the MC@NLO matched samples of the multijet-merged calculation are supplemented with exact NLO EW renormalised virtual corrections as well as approximated NLO EW real-emission corrections integrated over their real-emission phase space. This approximation is tailored to reproduce the exact NLO EW corrections in regions with large momentum transfers where they are dominated by virtual weak-boson exchanges and renormalisation corrections. The integrated-out real-photon emission part of the electroweak correction, which are of prime importance for leptonic final states, are recovered in a full event simulation by including a soft-photon resummation, *cf.* Sec. 2.6. Subleading tree-level contributions may be added where relevant.

The MEPS@NLO approach defines the current standard in simulating QCD-associated Standard Model production processes with SHERPA. Examples of validation and application of the method include:

- $V$+jets production with up to two jets described at NLO QCD and approximate NLO EW [143],

- $h$+jets production with up to three jets described at NLO QCD and 5 jets at LO [145, 146],

- four-lepton production [147],

- triple vector boson production [138],

- Higgs production in association with a gauge boson [138],

- application to loop-induced production processes [147, 148],

- and top-quark pair production in association with up to three jets [149].

In Sec. 3 we illustrate results for a variety of processes based on the matching and merging of matrix-element elements and parton showers and compare them with actual data from the LHC.

We close this section with an example for the process setup of Drell–Yan production in association with QCD jets, based on NLO QCD matrix elements for up to two jets:

```
(processes){
% process definition: Drell-Yan + 0,1,2 jets @ NLO QCD
Process 93 93 -> 11 -11 93{2};
Order (*,2);
% merging scale parameter corresponding to Qcut=30 GeV
CKKW sqr(30/E_CMS);
NLO_QCD_Mode MC@NLO;
RS_ME_Generator Comix;
Loop_Generator OpenLoops;
% include approx. EW corr. and first subleading tree-level
Associated_Contributions EW|LO1;
End process;
}(processes)
```

## 2.4  Internal Reweighting

The advancements of state-of-the-art QCD calculations as described in this publication led to a considerable growth in computational cost per event, a limiting factor in current and future

applications of event generators. One place where this cost can be addressed relatively easily is in studies targeting theory uncertainties for QCD input parameter and scale choices. Traditionally, this involved re-running the whole event-generation chain with different PDFs, values for the strong coupling $\alpha_s$, or with varied choices for the renormalisation and factorisation scales $\mu_{R,F}$. Nowadays this is achieved by appropriately reweighting the default prediction, significantly reducing the computational costs. Furthermore, SHERPA allows for a reweighting of the nominal NLO QCD calculation to include the associated approximate NLO EW corrections and subleading tree-level contributions.

**Implementation**    The parameter-reweighting techniques available in SHERPA have been described in [150]. Like in other generators, *cf.* [151, 152], they are calculated *on-the-fly* and cover scale variations, different PDF choices and modified values for coupling constants. They can furthermore include the effects these choices have on the parton shower, without rerunning it. The shower-emission reweighting uses the generalised Sudakov Veto Algorithm presented in [110]. Relative weights that emerge from different choices of these inputs are provided either in the HEPMC event output [37] or directly passed through the internal interface to the RIVET analysis framework [45]. Especially when the events are stored on disk, this also reduces the necessary disk space by potentially large factors, replacing full events for each variation by single numbers.

Reweighting in SHERPA can be applied to fixed-order calculations, both at LO and NLO using the NTUPLE decomposition [39]. When applied to matched or merged calculations, both the CSSHOWER and DIRE parton showers are supported. A lower bound on the parton-shower evolution scale can be set to omit the reweighting of very soft emissions. This allows for a trade-off between speed and accuracy.

An extensive example that invokes 7-point scale variations, variations over several PDF sets (including all their error replica/eigenvector PDFs, and sets with varied $\alpha_S(m_Z)$), and adding electroweak corrections as separate variations, is given by the following snippet:

```
(run){
% pairs of factors multiplying default squared scales muR,muF
SCALE_VARIATIONS 0.25,0.25 0.25,1. 1.,0.25 1.,1. 1.,4. 4.,1. 4.,4.;
% event variation for given (error) sets of PDFs
PDF_VARIATIONS CT14nlo[all] MMHT2014nlo68cl[all] NNPDF30_nlo_as_0118[all] \
NNPDF30_nlo_as_0115 NNPDF30_nlo_as0121;
% reweight nominal QCD to QCD+EW and QCD+EW+subLO
ASSOCIATED_CONTRIBUTIONS_VARIATIONS EW EW|LO1;
% enable consistent variations of parton-shower splittings
CSS_REWEIGHT 1;
% reweight the alpha_s that multiplies the splitting probability
REWEIGHT_SPLITTING_ALPHAS_SCALES 1;
% reweight the PDF ratios for initial-state splittings
REWEIGHT_SPLITTING_PDF_SCALES 1;
}(run)
```

Note that although the production will be considerably faster compared to producing separate event samples for each variation, the inclusion of hundreds of variations (as in the example above) can still slow down the production significantly, especially when enabling the parton-shower reweighting.

## 2.5   Initial State Radiation and PDFs

As a multi-purpose generator, SHERPA can be used to simulate collisions for various different collider setups, *e.g.* $pp$, $e^+e^-$, $ep$ or $\gamma\gamma$, or, more exotically, $\mu^+\mu^-$. This requires in particular

the proper modelling of beam spectra and (partonic) substructures.

**Beam Particles**   SHERPA allows for a two-step definition of particles entering a hard inter-
action: BEAM particles are specified, which may be subjected to a spectrum, modifying their
energy, or, possibly get converted to other particles, that are refered to as BUNCH. For the latter,
two examples are available in SHERPA, namely

- Laser Backscattering, where initial beam leptons are "converted" into bunch photons
  through Compton scattering [153–155]; and

- equivalent photons in the Weizsäcker–Williams approximation, where the beam particles
  act as quasi classical sources of collinear photon fluxes [156–158].

By default initial beams are considered monochromatic and will directly enter the second stage,
where their potential substructure is resolved.

**Encoding Partonic Structure: Available PDFs**   The emerging beam particles, that initiate
the hard scattering, may feature a partonic structure, described by a parton distribution func-
tion (PDF). This in particular applies to protons, photons, or leptons, whose constituents then
form the initial states of the matrix-element calculations described in Sec. 2.1.
For these beam particles SHERPA provides built-in PDFs that are shipped with the code, namely

- various proton PDFs, in particular the default set NNPDF 3.0 NNLO [159],

- the GRV leading-order photon PDF set [160, 161],

- and an analytic QED lepton structure function in different approximations [162–165].

In addition, SHERPA can be built with an interface to the LHAPDF library [166,167], allowing the
user ample choice in particular of proton PDFs, including their respective error and variational
sets.
An example beam setting, assuming proton-proton collisions at $\sqrt{s} = 13$ TeV using the MMHT
2014 NLO PDF set [168] via LHAPDF reads:

```
(run){
BEAM_1 2212; BEAM_ENERGY_1 6500.;
BEAM_2 2212; BEAM_ENERGY_2 6500.;

PDF_LIBRARY LHAPDFSherpa;
PDF_SET MMHT2014nlo68cl;
% use the PDF implementation of running aS
USE_PDF_ALPHAS 1;
}(run)
```

In setups which include PDFs, SHERPA automatically uses a consistent value of $\alpha_S$ and order
of its running throughout the event generation. When using LHAPDF, it is optionally possible
to use the actual implementation of the running within the given PDF library.

## 2.6 Higher-Order QED and EW Corrections to Decays

Higher-order QED and electroweak corrections can be computed in SHERPA using the soft-
photon resummation of Yennie, Frautschi and Suura (YFS) [169], which exploits the universal
structure of soft real and virtual photon emissions to construct an all-order approximation
while all mass effects are retained. The implementation in SHERPA [35,170] focusses on higher-
order corrections to particle decays, both for elementary particle decays (e.g. $W^\pm$, $Z$, $h$, $\tau^\pm$) as
well as for hadron decays.

**Implementation** The soft-photon resummed higher-order QED corrections in SHERPA are applied to decay processes that involve colourless particles only, while those that involve coloured particles – quarks and gluons – are subjected to a regular parton shower. By default, exact first-order QED corrections are applied to $Z \to \ell\ell$, $W \to \ell\nu$, $h \to \ell\ell$ and $\tau \to \ell\nu_\ell\nu_\tau$ and some hadron decays [35,103,146,171]. In all other cases, the eikonal approximation underlying the all-orders resummation is corrected in the hard collinear emission regime through subtracted Catani–Seymour dipole splitting functions [35].

**Treatment of Resonances** To meaningfully dress the complex final state of a hard scattering process with QED radiation it is mandatory to preserve its internal resonance structures. In SHERPA this is achieved through universal resonance identification described in [96]. It identifies all possible resonances by first scanning the final state of a scattering process for possible recombinations into resonant states present in the employed physics model. Then, all possible combinations are ordered by the difference of invariant mass of the decay products and the mass of the resonance, scaled by its width: $\Delta = |m_{\text{kin}}^{\text{inv}} - m_{\text{res}}|/\Gamma_{\text{res}}$. Resonances are identified in ascending order of $\Delta$, and configurations with $\Delta > \Delta_{\text{res}}$ are classified as a non-resonant production of the respective final state, where the arbitrary parameter $\Delta_{\text{res}}$ is set to 10 by default. The kinematics of the radiation off the thus identified resonant decay is subject to the condition that the invariant mass of the system is maintained. Non-resonantly produced final states are corrected for QED effects using the universal YFS exponential coupled with universal collinear-emission correction factors.

The main switches steering the YFS corrections are given by

```
(run){
% apply QED corrections to hard scattering - On/Off
ME_QED On;
% threshold \Delta_res to differentiate resonant and non-resonant regions
ME_QED_CLUSTERING_THRESHOLD 10.;
% general YFS switch: 0 - Off, 1 - soft photons only, 2 - soft and hard
   photons
YFS_MODE 2;
% apply exact first order QED matrix element corrections: 0 - Off, 1 - On
YFS_USE_ME 1;
}(run)
```

## 2.7 Underlying Event and Beam Remnants

The inner structure and finite size of incident hadrons in collisions, *e.g.* at the LHC, allow for effects beyond the hard process and secondary radiation. These are collectively called the underlying event (UE). In particular, partons inside the hadron may have some non-perturbative transverse momentum, and the break-up of the hadrons will produce further colour charges that will have an impact on the hadronisation of the partons. Furthermore, and maybe most prominently, it is possible to have more than one parton–parton interaction per hadron–hadron scatter. Such multiple parton interactions (MPIs) alter the overall particle yield in collisions, and they influence observables such as jet rates and jet shapes. The parameters introduced in the models addressing the underlying event are subject to tuning and need to be determined by comparing generator predictions to actual collider data, cf. Sec. 2.10.

**Modelling Multiple Parton Interactions** The first model successfully simulating MPIs as the dominant effect in the UE was proposed by Sjöstrand and van der Zijl in [31], and it is also the MPI model implemented in SHERPA. It is based on partonic $2 \to 2$ QCD scatters and the

observation that their cross section exceeds the total hadronic cross section even for moderate transverse momenta above $\sim$2–5 GeV. This is interpreted as having more than one parton–parton scatter per hadronic collision. The scatters are ordered by their transverse momentum, acting as an "evolution parameter" for the UE, which dresses the primary interaction with secondary scatters, through an expression similar to the Sudakov form factor in the parton shower:

$$P_{\text{no}}(p_{\perp,\text{min}}) = \exp\left( -\frac{1}{\xi \sigma_{\text{ND}}} \int\limits_{p_{\perp,\text{min}}^2} \mathrm{d}p_\perp^2 \frac{\mathrm{d}\hat{\sigma}}{\mathrm{d}p_\perp^2} \right), \tag{1}$$

where $\sigma_{\text{ND}}$ is the non-diffractive hadron–hadron cross section. Furthermore, $\hat{\sigma}$ denotes the parton-level $2 \to 2$ scattering cross section, including parton distribution functions, where the potential singular structure of the differential cross section, introduced by the $t$-channel singularity in the scattering amplitude at small momentum transfers and the divergent behaviour of the strong coupling at small scales, is tamed by supplementing the transverse momentum with a regulator $p_{\text{T},0}$, i.e. $p_\text{T}^2 \to p_\text{T}^2 + p_{\text{T},0}^2$. The evolution terminates when the transverse momentum of the secondary scatters falls below a cut-off value $p_{\perp,\text{min}}$, usually of the order of a few GeV. $\xi$ is a dimensionless parameter, allowing to rescale the non-diffractive cross section.

In their paper, Sjöstrand and van der Zijl also extended their model to describe Minimum Bias events; this, however, is not realised in SHERPA.

**Implementation** The Sjöstrand–van-der-Zijl model [31] has been implemented in SHERPA by precalculating and tabulating the partonic $2 \to 2$ scattering cross sections, using the results for the Sudakov-like factor driving the evolution of the MPIs in the transverse-momentum scale. These tables are either calculated and stored or read in during the initialisation phase of the run. SHERPA uses all partonic channels in MPIs, including processes with photons in the final state, and it supplements the scatters with a parton shower that starts at the transverse momentum of the scatter. The SHERPA implementation also features an impact-parameter dependence, given by the matter-density profile $\rho(r)$ of the incident hadrons. Available options are a simple Gaussian, an exponential, and, the default, a double Gaussian profile supporting a more compact matter core of radius $r_2$, containing the fraction $f_{\text{mat}}$ of the hadronic matter, surrounded by a larger sphere of radius $r_1$:

$$\rho(r) \propto (1 - f_{\text{mat}}) \frac{1}{r_1^3} \exp\left( -\frac{r^2}{r_1^2} \right) + f_{\text{mat}} \frac{1}{r_2^3} \exp\left( -\frac{r^2}{r_2^2} \right). \tag{2}$$

The corresponding profile parameters $f_{\text{mat}}$, $r_1$ and $r_2$, as well as the cut-off scale $p_{\perp,\text{min}}$, the regulator $p_{\text{T},0}$ and $\xi$ are subject of tuning to reference data. The SHERPA module for the underlying event is called AMISIC.

**Intrinsic Transverse Momentum** Partons inside hadrons are assigned a transverse momentum $k_\text{T}$ of the order of up to a few $\Lambda_{\text{QCD}}$. This is most visible for the case of Drell–Yan production of lepton pairs at small transverse momenta. There is a finite probability of the parton shower ending with no emissions down to its cut-off scale of about 1 GeV, which would lead to a visible peak at zero combined transverse momentum of the lepton pair. Instead the intrinsic $k_\text{T}$ washes out this unwanted and unphysical feature, and marginally shifts the overall distribution. In SHERPA, the intrinsic $k_\text{T}$ of partons in a beam hadron is chosen flavour- and $x$-independently according to a Gaussian distribution, parametrised by a mean value and a width:

$$\mathcal{P}(k_\perp) \propto \exp\left( -\frac{(k_\perp - \langle k_\perp \rangle)^2}{\sigma^2} \right). \tag{3}$$

It is applied to all partons stemming from the hadron break-up: the initiators of the parton shower at the cut-off scale for both the signal process and the MPIs as well as for all other partons that are added to guarantee flavour sum rules.

**Beam Remnants**   One subtlety in the modelling of the MPIs is the treatment of flavours and colours. For the former, flavour sum rules must be respected, which may necessitate to add extra quarks during the breakup of the hadron in the collision. Similarly, starting from a colour-neutral hadron it is clear that the colours of the partons must compensate each other, which offers some freedom in the colour assignments. In SHERPA this freedom is used to assign the colours such that the total length of the colour connections in momentum space, parametrised through the Lund measure [31, 172], is minimal.

For collisions that are not initiated by hadrons, the treatment of beam remnants is significantly less involved; in the case of initial-state radiation off leptons or similarly simple configurations, the beam remnant will be collinear to the incident beam but with reduced energy.

## 2.8   Hadronisation

There are currently two successful approaches included in event generators to describe the transition from the quanta of perturbative QCD, the quarks and gluons, to the observable hadrons, namely the Lund string model [33, 173] used in PYTHIA [1, 34] and cluster fragmentation models [174], such as the ones implemented in HERWIG [175] and SHERPA [32].

**Underlying Principles**   In both models, the parton configurations coming from the parton showers, underlying event and beam remnants are cast into the form of colour-connected singlets, which will decay non-perturbatively. These decays proceed by "popping" flavour/anti-flavour pairs and inserting them into the singlet structure, which in turn decays into more singlets with reduced masses. The only flavours being allowed to be produced in this way are the light $u$, $d$, and $s$ quarks and, possibly, diquarks made from them. The latter are hypothetical bound states of two quarks or two anti-quarks, forming a colour sextet or anti-sextet, which in the large-$N_c$ limit is re-interpreted as a colour anti-triplet or triplet. The diquarks also carry the baryonic quantum numbers – in this picture baryons are bound states of a quark and a diquark.

The Lund string and the cluster fragmentation models differ in the logic in which the non-perturbative flavour production proceeds. In the string model the singlets form coloured lines (strings) of the type $q g g \ldots g g \bar{q}$, which decay from their ends into a hadron and a "shorter" string. The flavour necessary to form the hadron is compensated by the anti-flavour of the new string end. In contrast, in the cluster model, gluons decay non-perturbatively to form colour-neutral quark/anti-quark or quark/diquark clusters. The clusters are interpreted as massive hadron resonances and undergo binary decays, until clusters are formed that are light enough to be hadrons.

**Cluster Fragmentation in AHADIC**   In SHERPA, the cluster model is implemented in the module AHADIC. It starts with non-perturbative gluon decays at the end of the perturbative phase which result in the production of quark/anti-quark and of diquark/anti-diquark pairs. In what follows the term "quark" is used such that it also includes diquarks. Their selection is driven by the phase space available for them, defined by their constituent masses, and by further flavour-specific suppression weights.

The splitting kinematics is realised in a dipole frame, where the gluons remain massless and the necessary recoil is provided by the spectator object. The splitting kinematics of the

gluon decays is defined by two parameters $y$ and $z$, with distributions given by

$$\mathcal{P}(y) \quad \propto \quad y^\eta \exp\left(-\frac{\left(y\hat{s} - \frac{y}{(1-y)}m^2_{\text{spect}}\right) - 4m^2_{\text{min}}}{4p^2_{\perp,0}}\right), \tag{4}$$

$$\mathcal{P}(z) \quad \propto \quad z^2 + (1-z)^2, \tag{5}$$

where $\hat{s}$ is the mass of the splitter–spectator system, $m_{\text{spect}}$ is the mass of the spectator, $m_{\text{min}}$ is a minimal mass of the resultant quark–anti-quark system, given by their constituent masses. The parameters $\eta$ and $p^2_{\perp,0}$ are subject to tuning, cf. Sec. 2.10.

The $y$-dependent term in the exponential denotes the invariant mass squared of the quark–anti-quark system, $m^2_{qq}$, which defines the allowed mass range of their flavours,

$$m^2_{qq} \equiv y\hat{s} - \frac{y}{1-y}m^2_{\text{spect}}. \tag{6}$$

In a centre-of-mass system, where the original gluon and spectator have momenta $p_A$ and $p_B$ oriented along the positive and negative $z$-axis, the momenta of the two quarks and the spectator after splitting are then given by

$$\begin{aligned}
p^\mu_q &= (1-z)(1-\beta)\cdot p^\mu_A &+& zy\cdot p^\mu_B &+& \vec{k}_\perp, \\
p^\mu_{\bar{q}} &= z(1-\beta)\cdot p^\mu_A &+& (1-z)y\cdot p^\mu_B &-& \vec{k}_\perp, \\
p^\mu_{\text{spect}} &= \beta\cdot p^\mu_A &+& (1-y)\cdot p^\mu_B,
\end{aligned} \tag{7}$$

where the terms involving $\beta = \frac{m^2_{\text{spect}}}{\hat{s}(1-y)}$ ensure that the spectator is on-shell. The transverse momentum $\vec{k}_\perp$ is distributed isotropically in the transverse plane. Its absolute value is given by

$$k^2_\perp = z(1-z)m^2_{qq} - m^2_q. \tag{8}$$

After the gluon decays, AHADIC proceeds with the formation of colour-neutral clusters, by combining colour-connected quarks and anti-quarks. Depending on their mass, these clusters either decay into hadrons or into further clusters. For both types of decays, flavour pairs have to be created again, using the same suppression weights as for gluon splittings.

For cluster decays into two clusters, a similar kinematics is built, with a new, non-perturbatively "popped" quark–anti-quark pair. Its mass is given by

$$m^2_{qq} = xy\hat{s}, \tag{9}$$

where $x$ and $y$ are the momentum fraction taken away from the splitter and the spectator partons within the decaying cluster, respectively. Their distributions are again given by

$$\mathcal{P}(x) = x^{\eta_x} P(m^2_{qq}) \quad \text{and} \quad \mathcal{P}(y) = y^{\eta_y} P(m^2_{qq}), \tag{10}$$

where

$$P(m^2_{qq}) \propto \exp\left(-\frac{m^2_{qq} - 4m^2_{\text{min}}}{4p^2_{\perp,0}}\right). \tag{11}$$

The exponents $\eta_{x,y}$ are determined depending on whether the quark associated to it is leading, *i.e.* has been produced perturbatively, or not, and whether it is the "splitter" or the "spectator", cf. Sec. 2.10.

To fix the kinematics of the new quark pair, again an additional energy-splitting variable $z$ is uniformly selected, such that the four-vectors of splitter, spectator, and new quarks are given by

$$
\begin{aligned}
p_q^\mu &= & zx \cdot p_A^\mu &+ & (1-z)y \cdot p_B^\mu &+ & k_\perp^\mu, \\
p_{\bar{q}}^\mu &= & (1-z)x \cdot p_A^\mu &+ & zy \cdot p_B^\mu &- & k_\perp^\mu, \\
p_{\text{spect}}^\mu &= & \alpha(1-x) \cdot p_A^\mu &+ & (1-\beta)(1-y) \cdot p_B^\mu, & & \\
p_{\text{split}}^\mu &= & (1-\alpha)(1-x) \cdot p_A^\mu &+ & \beta(1-y) \cdot p_B^\mu, & &
\end{aligned}
\tag{12}
$$

where $\alpha$ and $\beta$ are determined by the on-shell constraints of splitter and spectator, $p^2 = m^2$, and the squared transverse momentum is given by Eq. (8).

Clusters that are too light will decay into two hadrons; this is determined by comparing the cluster mass $M_c$ with a critical mass $M_{\text{crit}}$ determined by a combination of the masses of the lightest and heaviest hadron pairs, $M_-$ and $M_+$ that could emerge in the cluster decay,

$$
M_{\text{crit}} = M_-(1-\kappa) + M_+\kappa,
\tag{13}
$$

with an off-set parameter $\kappa$. If the cluster made of quarks $q_1\bar{q}_2$ is lighter than $M_c$ it will decay into two hadrons; the relative probabilities of an individual decay channel $C \to h_1 h_2$ is determined by a product of the "popping" probability of the necessary additional $q\bar{q}$ pair, $\mathcal{P}_q$, the flavour component of the wave function of the two hadrons, $|\psi_{1,2}|^2$, their meson or baryon multiplet weights, $\mathcal{P}_{\text{multi}}$, the decay phase-space weight, and a mass-dependent factor with parameter $\chi$,

$$
\begin{aligned}
\mathcal{P}(C \to h_1 h_2) = \mathcal{P}_q \, |\psi_1(q_1\bar{q})|^2 |\psi_2(q\bar{q}_2)|^2 \, \mathcal{P}_{\text{multi}} \, \sqrt{(M_c^2 - m_1^2 - m_2^2)^2 - 4m_1^2 m_2^2} \times \\
\times \, 8\pi M_c^2 \left( \frac{(m_1 + m_2)^2}{M_c^2} \right)^\chi.
\end{aligned}
\tag{14}
$$

Default values for the hadronisation parameters used to model gluon splittings, quark–anti-quark pair creation and cluster decays in SHERPA are compiled in App. A.

**Interface to Lund String Fragmentation** In addition to its native cluster model implementation, SHERPA also provides a link to the Lund string fragmentation model implemented in PYTHIA 6.4 [34]. The parameters of this model can be directly set through the run cards steering SHERPA.

## 2.9 Hadron Decays

Primary hadrons formed during the hadronisation stage are often unstable and will decay further into secondary hadrons. The same is also true for the $\tau$ lepton, which is unstable and predominantly decays into hadrons. Since decay products are often unstable themselves, a cascade of decays emerges.

**Organisation of Decay Chains** Hadron decays and their cascades in SHERPA are handled by its HADRONS module in a recursive approach, based on individual $1 \to n$ decays, first simulated assuming the incident hadron is on-shell. Spin correlations across the propagator of the decaying particle can be taken into account by the algorithm introduced in [36]. Off-shell kinematics is imposed a posteriori with a relativistic Breit–Wigner distribution, through the application of a reverse Rambo algorithm [105] which shifts the momenta to their new mass shells while preserving momentum conservation in the decay cascade. Since decaying particles have a finite lifetime they will travel in space before they decay and the resulting vertex offset is included in the simulation.

**Decay Widths and Kinematics**   Due to the plethora of observed hadron decay channels and the limited theoretical framework to predict them precisely, the decay tables are based on measured branching ratios [176]. For some particles the branching ratios of observed decays do not add up to unity. In such cases, the branching ratios are rescaled within their known uncertainties to add up to one. It is also possible that the known decay modes are not sufficient – this is particularly true for heavy mesons and baryons. For them the known decay tables are amended with partonic decays of one of the constituent quarks with subsequent parton showering and hadronisation. This could of course lead to an exclusive final state already present in the decay table – in such a case the resulting hadronic final state is vetoed and the procedure repeated, until a legitimate final state is produced.

The kinematics of each decay step is generated according to generic matrix elements representing the spin structure of the involved particles. Furthermore, in many cases and in particular for weak decays involving hadrons, a wide variety of form-factor models are implemented, thus parametrising the weak decay of quarks in a bound state beyond the generic spin matrix elements. State-of-the-art decay tables and form-factor implementations are provided *e.g.* for decays of the $\tau$, $B^0$, $B^{\pm}$, $B_s$, $B_c^{\pm}$, $D^0$, $D^{\pm}$, $D_s$, $\Lambda_b$, $\Lambda_c^{\pm}$.

## 2.10   Tuning non-perturbative model parameters

The non-perturbative models used to address the parton-to-hadron transition, the intrinsic motion of partons bound in composite initial states and the underlying event involve a number of parameters, that are not determined by first principles. Rather, they need to be adjusted through an iterative comparison of corresponding SHERPA predictions with experimental data. This tuning procedure is achieved in three consecutive steps, implicitly assuming that the respective phases of the event generation factorise sufficiently. Input to these models are well-defined perturbative matrix-element calculations with parton showers attached, that, in turn, evolve the hard-process particles into a parton ensemble with minimal inter-parton separations of order the parton-shower cut-off scale, independent of the hard-process momentum transfer. These perturbative calculations are specified by a set of input parameters, that also affect the subsequent non-perturbative evolution. Most importantly the strong coupling $\alpha_s$ and the parton density functions. Per default, in SHERPA, the value and the running of $\alpha_s$ is set in accordance with the PDFs employed. The standard PDF set of SHERPA is NNPDF 3.0 NNLO [159], with $\alpha_s(M_Z) = 0.118$ and two-loop QCD running. For leptonic initial states we assume $\alpha_s(M_Z) = 0.118$ and use a two-loop running as well.

**Tuning of the cluster hadronisation**   In a first step, the parameters of the cluster hadronisation are tuned, usually with respect to data from LEP 1, such as

- the mean and distribution of the charged-particle multiplicity;

- the yields of individual hadron species, in particular charged and neutral pions and kaons, protons, lambdas, heavy mesons and baryons;

- the distribution of charged particles in phase space with respect to the thrust axis, *i.e.* their rapidities and transverse momenta inside and outside the event plane;

- the fragmentation function of $B$ hadrons; and

- event shapes, and especially thrust, thrust major and minor as well as differential jet rates.

Given that the SHERPA cluster fragmentation model features about 20 parameters to describe the non-perturbative splitting of final-state gluons, the formation of mesonic and baryonic

clusters, their subsequent decays and ultimately the creation of primary hadrons, cf. Sec. 2.8, their tuning procedure is largely automated. It relies on the use of the PROFESSOR tuning tool [177] to optimise the description of the reference data. The resulting main parameters of the SHERPA cluster-hadronisation model are compiled in App. A. Assuming universality of the hadronisation model these parameters are kept fixed also for other collision energies and beam particles.

**Tuning of the intrinsic transverse momentum**    Assuming an incoming proton beam two parameters determine the intrinsic transverse-momentum distribution of its constituents, *i.e.* the mean and width of the hypothesised Gaussian distribution, cf. Eq. (3). These are adjusted by studying the transverse-momentum distribution of Drell–Yan lepton pairs in proton–proton collisions at $\sqrt{s} \equiv E_{\text{ref}} = 7$ TeV. We thereby assume identical parameter values for the two proton beams. The current default tune results $\langle k_\perp \rangle = 1.1$ GeV and $\sigma = 0.85$ GeV. The determined width parameter is scaled to other centre-of-mass energies according to

$$\sigma(E_{\text{cms}}) = \sigma(E_{\text{ref}}) \left( \frac{E_{\text{cms}}}{E_{\text{ref}}} \right)^{0.55}. \tag{15}$$

**Tuning of the underlying event**    With the parameters of the cluster hadronisation and the intrinsic transverse-momentum adjusted, the tuning of the underlying-event model remains. To this end the relevant quantities of the impact-parameter dependent multiple-parton-interaction model, cf. Sec. 2.7, are adjusted. We use the PROFESSOR tuning tool for this task and employ dedicated measurements from Tevatron and LHC as reference data. The obtained default set of model parameters is again summarised in App. A. Note, the quoted values for $p_{T,\text{min}}$ and $p_{T,0}$ are for a reference collision energy of $\sqrt{s} \equiv E_{\text{ref}} = 1.8$ TeV. They get evolved to the actual collider energy $E_{\text{cms}}$ according to

$$p_{T,i}(E_{\text{cms}}) = p_{T,i}(E_{\text{ref}}) \left( \frac{E_{\text{cms}}}{E_{\text{ref}}} \right)^{\alpha}, \tag{16}$$

with the power $\alpha$ set to 0.244.

# 3 Highlighting SHERPA Applications

In this section we present selected results obtained with releases of the SHERPA-2 series. The purpose mainly lies in giving illustrative applications of the calculational methods and physics models introduced in Sec. 2. While presenting these examples, we will highlight specific features and aspects of the simulation chain. Where available, we directly compare to experimental data, gauging the quality of our predictions. In fact, in many cases SHERPA has been used in the actual analysis of the data providing state-of-the-art signal and background samples, being vital for the proper interpretation of the measurements.

Sections 3.1–3.8 focus on aspects of the combination of QCD matrix elements and parton showers when applied to processes such as jet-associated vector- and Higgs-boson production, top-quark single and pair production, or vector-boson pair creation, including channels with photons. Sec. 3.9 is devoted to simulations of physics beyond the Standard Model, while Sec. 3.10 focuses on non-perturbative aspects of the simulation: the hadronisation and hadron decays, as well as the underlying event in proton-proton collisions. If not stated explicitly, in all applications presented here the default SHERPA tune, *i.e.* the respective set of non-perturbative model parameters listed in App. A, has been used.

### 3.1  $Z(\to \ell\ell)$ production in association with jets

We begin the discussion with the most prominent testbed for calculational schemes combining QCD matrix elements with parton showers, namely the production of a massive vector boson in association with jets. These processes feature significant production rates at the LHC and probe a wide range of kinematic configurations, from almost exclusive vector-boson production to signatures featuring very hard jets and a gauge boson at a rather small transverse momentum. While studies of these scenarios are interesting on their own, they reflect specific situations where $V$+jets production has to be considered as an important Standard Model background in New Physics searches. Accordingly, a realistic simulation needs to address not only the jet-production rates, but also their distributions in the bulk and the tails of various observables. Furthermore, kinematic correlations between the final-state objects need to be modelled correctly.

In Fig. 2 we present a few results for $Z$+jets production in proton-proton collisions at $\sqrt{s} = 13$ TeV with off-shell decays for $Z \to \ell\ell$ and compare the SHERPA predictions with data from ATLAS [178]. We refer the reader to [178] which describes the event-selection criteria used. Figs. 2a and 2b show the distribution for the number of jets, $N_{\text{jets}}$, and the azimuthal correlation between the two leading jets, *i.e.* $\Delta\phi(j_1, j_2)$.[3] While the $N_{\text{jets}}$ distribution probes multijet production rates, $\Delta\phi(j_1, j_2)$ is sensitive to kinematic correlations between jet momenta. Figures 2c and 2d show the scalar sum of jet and lepton transverse momenta, commonly referred to as $H_{\text{T}}$, and the invariant mass of the pair of leading jets $M(j_1, j_2)$. $H_{\text{T}}$ is sensitive to the $p_{\text{T}}$ spectra of the leading jets. The tail of the distribution probes higher multiplicities, and can therefore not be described by the parton shower alone. The invariant mass distribution is sensitive to non-perturbative effects at small values and forms an important background for New-Physics searches at large values.

The SHERPA prediction in these plots is obtained from multijet merging, applying the MEPS@NLO method described in Section 2.3. In practice, we consider matrix elements for the production of an electron anti-electron pair with zero, one and two jets computed at NLO accuracy in the strong coupling, matched to the parton shower with the MC@NLO prescription, while the $Z$+3- and $Z$+4-jets calculations are included at LO only. The merging cut parameter is set to $Q_{\text{cut}} = 20$ GeV. The parton showered events are hadronised by the cluster fragmentation and the underlying event is simulated through the AMISIC module. QED corrections are enabled for the leptonic decay of the intermediate $\gamma^*/Z$, *cf.* Sec. 2.6. The scale-variation band shown in Figs. 2a and 2d is obtained through the "on-the-fly" reweighting described in Section 2.4 for a 7-point scale variation with factors of 1/2 and 2, including the scale dependence of both the fixed-order and the parton-shower calculation in a consistent way. We further include approximate NLO EW corrections. These, however, have negligible impact for the considered observables.

In the SHERPA calculation up to four jets might be seeded by hard matrix-element partons, jet multiplicities beyond that originate from the parton shower. We observe that the $N_{\text{jets}}$ distribution is well modelled by SHERPA even up to six jets. The azimuthal correlation between the two hardest jets is well described by the MEPS@NLO method, in contrast to calculations where the first and/or second jet originate from a spin-averaged parton-shower emission. The $H_{\text{T}}$ distribution, shown in Fig. 2c, is well described by SHERPA as is the invariant mass of the two leading jets, depicted in Fig. 2d.

---

[3]Note that a jet is called "leading" if it is the one with the highest transverse momentum, and that the jets $j_i$ ($i = 1, \ldots, N_{\text{jets}}$) are ordered descending in their transverse momentum.

Figure 2: Results for various observables in $Z$+jets production at the LHC. The uncertainty bands for the SHERPA predictions correspond to the envelope over a 7-point scale variation, whereas their error bars indicate the Monte-Carlo error. In addition, the effect of adding approximate electro-weak corrections to the nominal predictions is shown.

(a) inclusive jet cross sections

(b) azimuthal correlation of the two leading jets

(c) scalar sum of the dilepton transverse mass and the jet transverse momenta

(d) invariant mass of the two leading jets

## 3.2 $W(\rightarrow \ell\nu)$ production in association with jets

We proceed with the inclusive production of a leptonically decaying $W$ boson. Besides the importance of incorporating higher-order QCD matrix elements in the simulation, we illustrate

the impact of electroweak one-loop corrections. In Fig. 3 we present the gauge-boson transverse momentum distribution in proton-proton collisions at $\sqrt{s} = 13$ TeV evaluated in various approximations. The $W$ boson is reconstructed from the charged lepton and the missing transverse momentum, where only modest acceptance cuts are applied.

The standard MEPS@NLO QCD prediction is contrasted with its LO variant MEPS@LO. In both calculations matrix elements with up to two jets, at NLO QCD and LO QCD accuracy, respectively, have been matched to the parton shower and merged into an inclusive sample. For the merging criterion we chose $Q_{\mathrm{cut}} = 20$ GeV. Further details on the calculational setups can be found in Ref. [143]. Comparing the blue and green uncertainty bands, it is apparent that the prediction based on exact NLO QCD matrix elements features a significantly reduced theoretical uncertainty.

Including approximate NLO EW corrections in MEPS@NLO QCD+EW$_{\mathrm{approx}}$, *cf.* Sec. 2.3, has an important impact for $W$ production at large transverse momenta, exhibiting the familiar structure of the well-known EW Sudakov suppression. The corresponding one-loop virtual amplitudes for up to $W + 2j$ production have been obtained from OPENLOOPS. Subleading mixed QCD-EW tree-level contributions are provided by COMIX. Their impact is very marginal on this observable, however, this is different for the leading-jet transverse-momentum distribution, *cf.* [143].

### 3.3 $gg \to h$ production in association with jets

Higgs-boson production processes form a centre piece of the LHC physics program. This is in particular true for the case of the gluon-fusion channel, as it features the largest cross section. It is commonly described in the Higgs Effective Field Theory (HEFT) approach. In the complete Standard Model it constitutes a loop-induced process, with top and bottom quarks propagating. With SHERPA both approaches can be employed.

We present results based on inclusive Higgs production as well as Higgs production in association with one jet at NLO accuracy in the strong coupling, while Higgs production in association with two and three jets is described at LO accuracy, merged using the standard MEPS@NLO method, *cf.* [145]. While the HEFT computation proceeds straight-forwardly, the full Standard Model computation reweights each component of the NLO calculation with its loop-induced counter-part [179]. Only the virtual corrections, which are structurally of two-loop origin including different and dynamic mass scales, and have only been calculated recently [180], are approximated by factorising the NLO correction in the effective theory and the mass corrections at LO. This approximation has been shown to well reproduce the shape of the full NLO results for the Higgs-boson $p_{\mathrm{T}}$ distribution [180].

Fig. 4 details the 0-, 1-, 2- and 3-jet inclusive distributions of the transverse momentum of the Higgs boson in gluon-fusion production. It is interesting to see, that the quark-mass corrections introduced through the top-quark running in the loop in the exact Standard Model calculation, are independent of the number of jets that are accompanying the Higgs boson. This mass suppression reaches up to $-60\%$ at transverse momenta of around 500 GeV and increases further towards higher $p_{\mathrm{T}}$. The finite mass corrections in dependence on the jet multiplicity have been further investigated in [88]. With the exception of the scalar sum of transverse momenta, $H_{\mathrm{T}}$, no appreciable dependence was found among the observables investigated, both in inclusive Higgs boson production and in Higgs boson production through gluon fusion in VBF kinematics.

For completeness we list the following additional flags which instruct SHERPA to perform the outlined reweighting procedure using the appropriate loop-induced processes from the OPENLOOPS library:

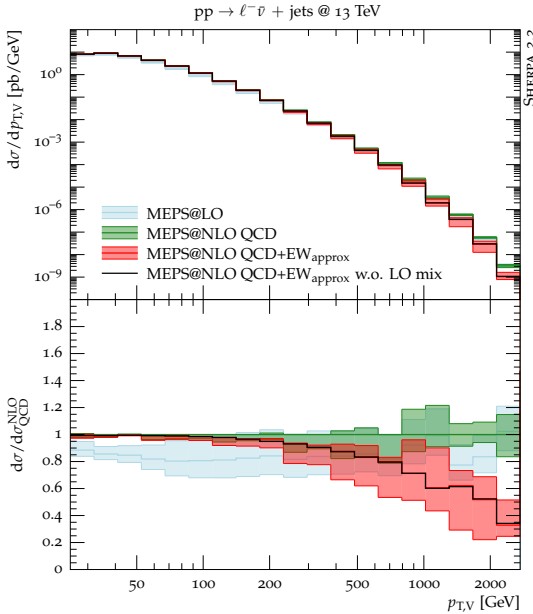

Figure 3: Predictions for the $W$-boson transverse momentum distribution at the LHC.

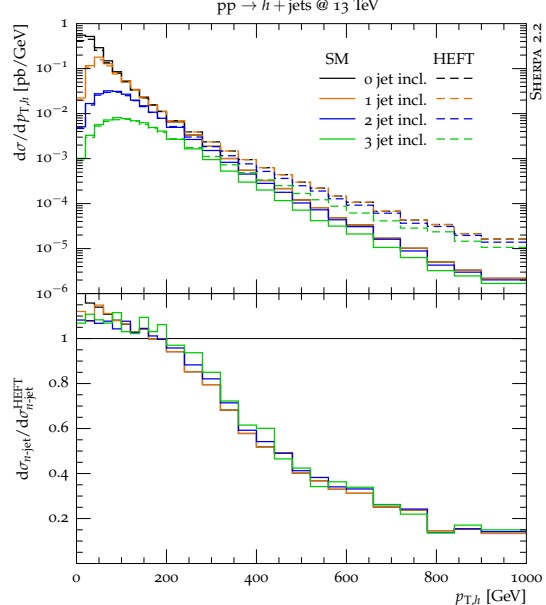

Figure 4: Predictions for the Higgs-boson transverse momentum distribution in gluon-fusion production at the LHC.

```
(run){
% finite top mass effects
KFACTOR GGH;
OL_IGNORE_MODEL 1;
OL_PARAMETERS preset 2 allowed_libs pph2,pphj2,pphjj2 psp_tolerance 1.0e-7;
}(run);
```

### 3.4 $t\bar{t}$ production in association with jets

The production of a top-quark pair in proton-proton collisions is particularly challenging due to the non-negligible mass and finite life-time of the colour-charged tops.

Fig. 5 shows the visible energy ($H_T$) distribution in top-quark pair production as predicted by the NLO multijet merging in SHERPA. This calculation involves NLO fixed-order input predictions with up to two light jets in addition to the top-quark pair. The top-quark decays are calculated at leading order including spin correlations based on the $t\bar{t}$+jets Born matrix elements using spin-density matrices, *cf.* Sec. 2.1. The one-loop matrix elements were obtained from OPENLOOPS. The MC@NLO matching for heavy quarks applied in the simulation is based on the massive Catani–Seymour dipole subtraction [10] and was originally constructed in [181]. Further details on the calculational setup can be found in [149].

Besides the MEPS@NLO result we present the corresponding MEPS@LO prediction. Note the excellent agreement between the two predictions, after the leading-order result has been multiplied by a global $K$-factor of 1.65. The first ratio panel in Fig. 5 shows clearly, that, beyond this global $K$-factor, the main effect of the higher-order corrections is a drastic reduction of the scale uncertainty, which in this case has been determined by varying the renormalisation and factorisation scales, but not the resummation scale. The second lower panel shows the individual contributions of $t\bar{t}$ (solid), $t\bar{t}j$ (dash-dotted) and $t\bar{t}jj$ (dotted) final states to the

overall result. At low $H_T$ all components contribute to the overall result, while at high $H_T$ the prediction is given almost entirely by the $t\bar{t}jj$ component.

## 3.5 Single-top quark production

In Ref. [123] a dedicated SHERPA study of single-top quark production in hadronic collisions has been presented which is challenging due to the various production modes and their differing characteristics in how the final-state phase space is populated. Our study includes the consistent evaluation in the four- and five-flavour PDF schemes and process-definition ambiguities when considering higher-order corrections, where a separation from top-quark pair production has to be defined. With SHERPA single-top quark production in the $s$, $t$ and $tW$ channels can be simulated using the MC@NLO implementation.

In Fig. 6, we compare MC@NLO results for the reconstructed top-quark transverse-momentum distribution in the $t$-channel production mode in the four- and five-flavour scheme with ATLAS data taken at $\sqrt{s} = 8$ TeV [182]. The bands correspond to the theory error convention used in [182]. That is, the statistical, the strong coupling, the PDF and the (dominant) 7-point scale uncertainties, all added in quadrature. The SHERPA predictions and experimental data agree within their respective uncertainties. For further details on the calculation and additional results, see [123]. The minimal settings to generate $t$-channel single-top production events at MC@NLO with SHERPA are:

```
(run){
% single-top specific scale definition
CORE_SCALE SingleTop;
% enable decays of produced top-quarks
HARD_DECAYS On;
...
}(run)
(processes){
Process 93 93 -> 6 93;
Order (*,2); NLO_QCD_Mode MC@NLO;
% require t-channel propagator
Min_N_TChannels 1;
...
End process;
}(processes)
```

## 3.6 Diboson production in association with jets

Another important class of benchmark processes at hadron colliders is diboson production. This includes the pair production of massive gauge bosons, *i.e.* $W$ and $Z$, but also photon pairs or mixed $W\gamma$, $Z\gamma$ final states. These channels provide precision tests of the electroweak sector, including triple- and even quartic gauge couplings. They form irreducible backgrounds for Higgs-boson production with the Higgs decaying into gauge bosons, or searches for New Physics. Besides the accurate modelling of the diboson final states, a realistic description of the associated QCD activity is vital, as it often provides the only handle to separate signal from irreducible backgrounds. All the channels mentioned above have loop-induced contributions such as $gg \to W^+W^-$, that are phenomenologically important and require refinements in the techniques for combining matrix elements with parton showers.

We begin the discussion with one of the main Higgs-production backgrounds: $pp \to WW^*$, *i.e.* $pp \to \ell\nu\ell\nu$, with two charged leptons of different flavour and the corresponding neutrinos. A dedicated analysis with SHERPA has been presented in [147]. In Fig. 7 we present the leading-

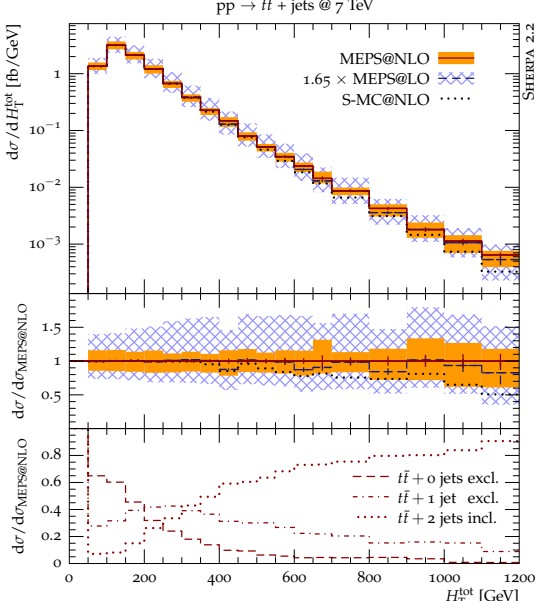

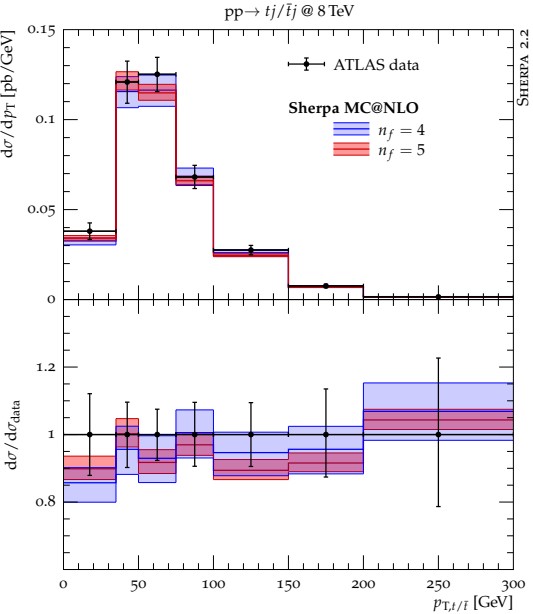

Figure 5: Predictions for the $H_T$ distribution in top-quark pair production at the LHC.

Figure 6: Results for the $p_T$ distribution of reconstructed top-quarks in $t$-channel single-top production at the LHC. The four- and five-flavour MC@NLO are compared with data from [182].

jet $p_T$ distribution for this off-shell diboson-production channel. The upper panel shows the SHERPA MEPS@NLO prediction when merging the zero- and one-jet contributions, with the QCD one-loop matrix elements provided by OPENLOOPS [14]. The uncertainty bands correspond to the perturbative (red) and resummation (blue) scale variations, again added in quadrature to yield an overall uncertainty estimate (yellow band).

In the first ratio plot the MEPS@NLO prediction is compared to an inclusive MC@NLO (red dashed) calculation, based on the four-lepton ($4\ell$) NLO QCD matrix element matched to the SHERPA parton shower. Further, we present the pure fixed-order result based on the $4\ell + 1$jet NLO QCD matrix element (blue dashed). Note that the inclusive MC@NLO prediction describes this observable only at LO precision, and is found not to be compatible with the more precise MEPS@NLO prediction over a wide range of the spectrum. Details on the simulation setups and parameters used can be found in [147].

The lower panel displays the relative corrections and uncertainties of a multijet-merged prediction of loop-induced $gg \rightarrow \ell\nu\ell\nu$ production in association with jets, dubbed MEPS@LOOP$^2$, normalised to MEPS@NLO at the central scale. These squared quark-loop amplitudes constitute higher-order corrections to the generic $4\ell$ and $4\ell + 1$jet processes. However, their relative contribution can be as large as 5% around $p_T(j_1) \approx 20$ GeV.

A more detailed view on these loop-induced corrections is provided in Fig. 8. Here the multijet-merged sample is compared to a simple LOPS prediction of $gg \rightarrow \ell\nu\ell\nu$ production, dubbed LOOP$^2$PS here. Furthermore, the contributions of the $4\ell + 0j$ and $4\ell + 1j$ matrix element to the full MEPS@NLO sample are indicated. It is evident that at high $p_T$ the relevant contributions are those of the one-jet process, which can not be fully accounted for by the pure parton shower in the LOOP$^2$PS sample.

Note, a very recent experimental measurement of this channel at $\sqrt{s} = 13$ TeV, including an extensive comparison of state-of-the-art theoretical predictions with data, among them those

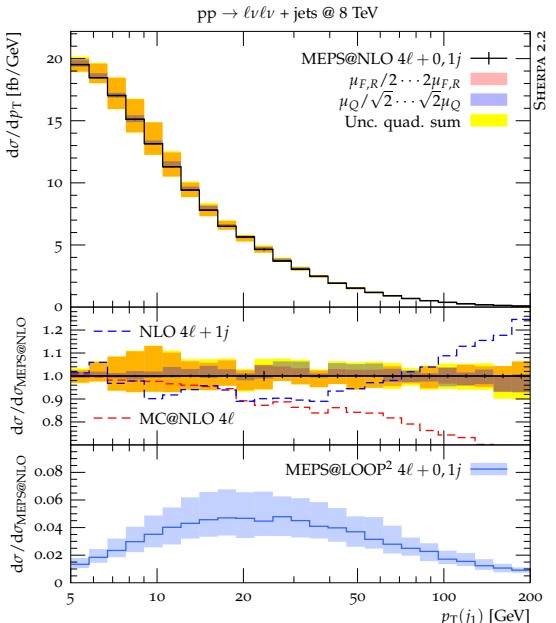

Figure 7: Prediction for the leading jet transverse-momentum distribution in $pp \to \ell\nu\ell\nu$ production in association with jets at the LHC.

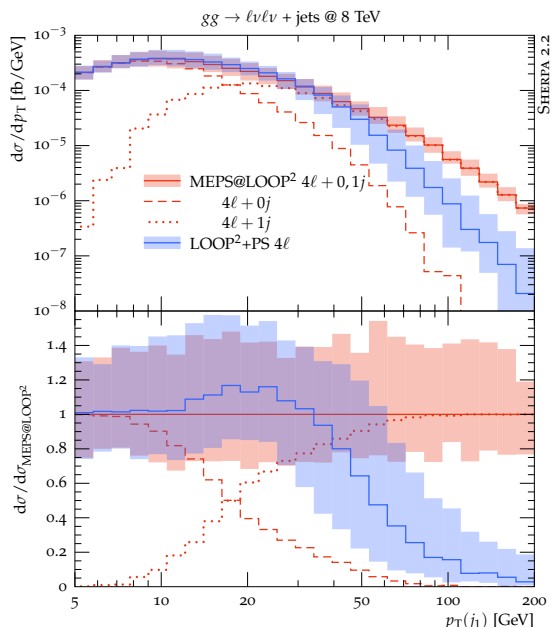

Figure 8: Predictions for the leading jet transverse-momentum distribution in loop-induced $pp \to \ell\nu\ell\nu$ production in association with jets at the LHC.

from SHERPA, has been presented by the ATLAS collaboration in [183]. A similar study but for the final state of four charged leptons has been presented in [184].

### 3.7 $V\gamma$ production in association with jets

The second diboson channel we want to discuss here is the associated production of a prompt photon and a lepton-pair, possibly accompanied by additional QCD jets. The corresponding study has been presented in [185], which we refer to for details on the generator setup, parameter choices and object definitions.

MEPS@NLO predictions for the transverse-momentum distribution of the photon based on merging of $pp \to e^+e^-\gamma + 0, 1$ jets@NLO $+ 2, 3$ jets@LO each matched to the parton shower is presented in Fig. 9. The prediction is compared with an ATLAS measurement [186] at $\sqrt{s} = 8$ TeV. Furthermore, results based on MEPS@LO and an inclusive MC@NLO simulation are shown.

Most notably, the MEPS@NLO calculation is in very good agreement with the data, both in rate and shape over the whole range of the observable. It is interesting to note that, similar to Sec. 3.4, the MEPS@LO prediction largely agrees in shape with the NLO merged one as can be seen in the upper ratio panel. The effect of going from LO to NLO accuracy in the simulation can be captured by a global $K$-factor which brings the central prediction in good agreement with experimental data. More importantly, NLO accurate predictions show significantly reduced inherent uncertainties, which are estimated by variations of the perturbative scales and PDFs, see the lower two ratio panels.

## 3.8 Diphoton production in association with jets

Predictions for prompt-photon production are notoriously difficult, especially for low-energetic or not well isolated photons. Appropriate choices for the perturbative scales need to be made that are valid for a wide range of kinematics and, potentially, non-perturbative contributions need to be considered. In particular, a fragmentation component has to be taken into account, where soft or collinear photons are emitted from harder jets through QED $q \to q\gamma$ splittings. One option to do so is a combined QCD ⊗ QED parton shower and related multijet merging, as proposed in [110]. As an implementation of such an algorithm is not available at NLO accuracy yet, we use a QCD MEPS@NLO setup here, but take fragmentation-like configurations of a hard jet and a soft photon into account through higher-multiplicity matrix elements. To make the fragmentation component as inclusive as possible, we use a dynamic merging cut [115] with $\bar{Q}_{cut} = 10$ GeV using the following run-parameter settings:

```
(run){
% core scale m_yy
CORE_SCALE VAR{Abs2(p[2]+p[3])};
...
}(run)
(processes){
Process 93 93 -> 22 22 93{3};
Order (*,2);
% dynamical merging cut with Qcutbar=10.0 GeV and mu=m_yy
CKKW sqr(10.0/E_CMS)/(1.0+sqr(10.0/0.6)/Abs2(p[2]+p[3]));
...
End process;
}(processes)
```

To mitigate the mismatch of the photon-isolation cuts between the generator level and the experimental analysis, we choose a hybrid isolation approach as described in more details in [187].

Accordingly, NLOPs matched simulations for $pp \to \gamma\gamma$ and $pp \to \gamma\gamma$+jet production are merged into an inclusive sample and additionally, matrix elements with up to three partons in the final state are included at LO accuracy in the approach of [29]. The comparison with data from ATLAS [188] for the transverse-momentum distribution for diphoton production in Fig. 10 shows good agreement in all regions of the spectrum. Note, the CMS collaboration also presented an analysis of diphoton production at $\sqrt{s} = 7$ TeV [189], where MEPS@LO predictions from SHERPA provided a very good description of the data.

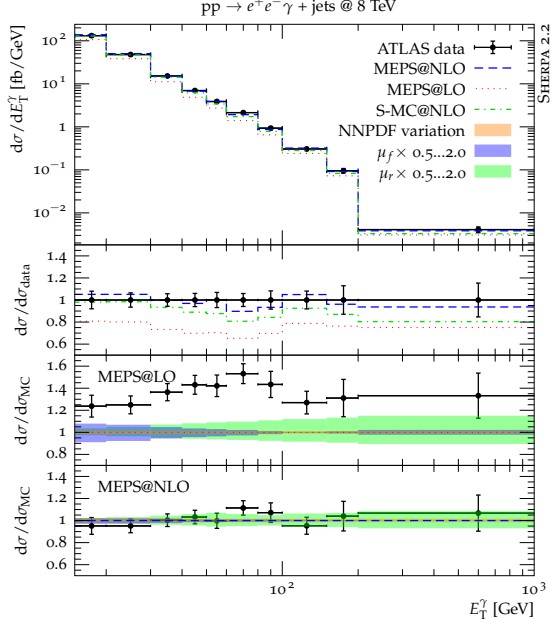

Figure 9: Results for the photon transverse-momentum distribution in $pp \rightarrow e^+e^-\gamma$ production in association with jets at the LHC, comparing to data from [186].

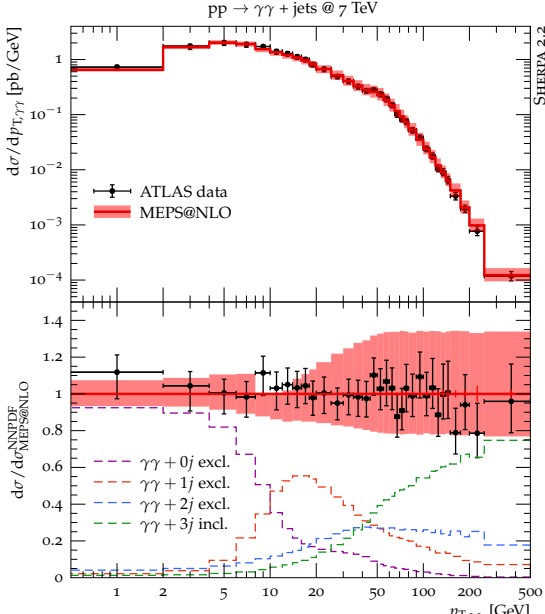

Figure 10: Result for the diphoton transverse-momentum distribution at the LHC, in comparison to data taken from [188].

### 3.9 Physics beyond the Standard Model

We now present two examples in which SHERPA is used as a generator for a New Physics signal. First, an analysis of dimension-six gluon operators in multijet production at a Future Circular Hadron Collider (FCC) with $\sqrt{s} = 100$ TeV. Further details on this study can be found in [67, 190]. Second, a study on an anomalous triple gauge coupling in $Z$-boson pair production at the LHC, based on the corresponding CMS measurement [191].

A study presented in [67] considers the impact of additional dimension-six gluon interactions given by the effective operator

$$c_G \mathcal{O}_G = \frac{g_s \, c_G}{\Lambda^2} f_{abc} G_{a\nu}^{\rho} G_{b\lambda}^{\nu} G_{c\rho}^{\lambda} \quad \text{with} \quad G_a^{\rho\nu} = \partial^\rho G_a^\nu - \partial^\nu G_a^\rho - i g_s f_{abc} G^{b\rho} G^{c\nu}, \tag{17}$$

on multijet production at the LHC. The corresponding model, which needs to be invoked by SHERPA, has been obtained through a FEYNRULES implementation of the interactions, subsequently interfaced to SHERPA using the UFO standard, as described in Section 2.1. The matrix element generator COMIX has been used to evaluate all contributing Lorentz and $SU(3)$ colour structures [8]. For SM backgrounds as well as for the signal (which interferes with the SM amplitudes) corresponding leading-order matrix elements for up to five jets are merged via the MEPS@LO method described in Section 2.3.

In Fig. 11 we show the effect on the $S_T$ distribution, with or without the contributions from Eq. (17) for a selection of inclusive five-jet events at FCC energies, where $S_T$ denotes the scalar sum of the transverse momentum of all reconstructed jets, with $p_{T,j} > 1$ TeV and $|\eta_j| < 5.2$. Here the relevant ratio of the scale $\Lambda$ and the Wilson coefficient $c_G$ is taken to be $\Lambda/\sqrt{c_G} = 50$ TeV. For the considered luminosity of 10 ab$^{-1}$ the New Physics signal exceeds the given uncertainty band of the SM prediction, based on variations of the perturbative scales, at around $S_T \gtrsim 40$ TeV.

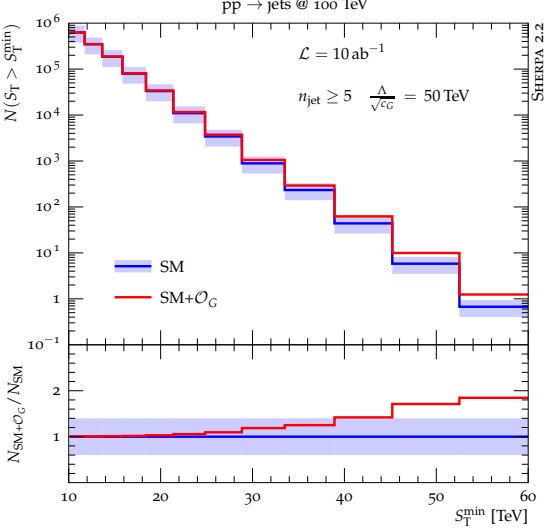

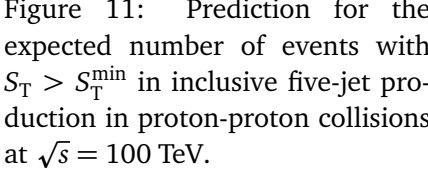

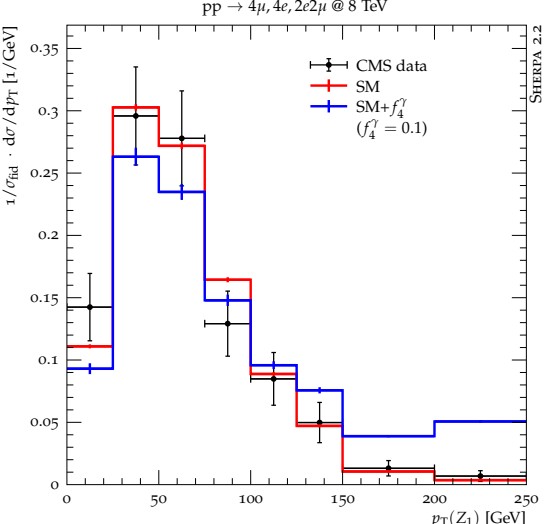

Figure 11: Prediction for the expected number of events with $S_T > S_T^{\min}$ in inclusive five-jet production in proton-proton collisions at $\sqrt{s} = 100$ TeV.

Figure 12: Results for the transverse momentum of the $Z$-boson candidate closest to the nominal $Z$-boson mass in four-lepton events at the LHC, comparing to data taken from Ref. [191].

The second example is related to anomalous triple gauge couplings in the electroweak sector of the Standard Model. For this we prepared a FEYNRULES implementation of the general $WWV$ and $ZZV$ Lagrangian considered in [192], where $V$ denotes either a $Z$-boson or a photon. This theory features for example a CP-violating $ZZ\gamma$ coupling, proportional to the form factor $f_4^\gamma$, where it is assumed that the two $Z$ bosons are on-shell. The best testbed for this type of interaction is $Z$-boson pair production. In Ref. [191] the CMS collaboration reported on a corresponding search for anomalous $ZZZ$ and $ZZ\gamma$ interactions in four-lepton production in 8 TeV proton-proton collision events. The final states $4e$, $4\mu$ and $2e2\mu$ are taken into account. The event-selection criteria used read

$$p_T(\mu) > 5 \text{ GeV}, \quad p_T(e) > 7 \text{ GeV}, \quad |\eta(\mu)| < 2.4, |\eta(e)| < 2.5$$
$$\text{and} \quad m_{e^+e^-/\mu^+\mu^-} \in [60, 120] \text{ GeV}. \tag{18}$$

In the experimental analysis the SHERPA generator has been used for the signal predictions.

In Fig. 12 we compare leading-order plus parton-shower predictions from SHERPA with CMS data published in [191]. Besides the leading-order SM expectation we show as an illustrative example the prediction when including a $ZZ\gamma$ vertex with coupling $f_4^\gamma = 0.1$, with all other New Physics couplings set to zero. Clearly, the latter hypothesis is not compatible with the observed data. The CMS collaboration extracted 95% confidence level limits on $f_4^\gamma \in [-0.005, 0.005]$.

## 3.10 Hadronisation, Underlying Events and Hadron Decays

This section is devoted to highlight some aspects of the modelling of non-perturbative phenomena in SHERPA. In particular, we present results sensitive to hadronisation, the underlying event and (soft) hadron decays, including spin correlations in hadronic $\tau$-decays.

**Hadronisation** SHERPA implements a cluster model for the fragmentation of partons into hadrons, *cf.* Sec. 2.8 and Ref. [32]. Furthermore, it offers an interface to the Lund string

fragmentation model as implemented in PYTHIA 6.4 [34]. This allows for important cross checks of the non-perturbative modelling. In particular it is possible to extract theoretical uncertainties related to the parton-to-hadron transition, keeping all perturbative aspects of the simulation identical.

To illustrate this aspect, we show a comparison with LEP data from ALEPH [193] for the thrust and total jet broadening event-shape variables in Fig. 13 and Fig. 14, respectively. The SHERPA predictions presented there are based on an MEPS@NLO sample, where the $2 \rightarrow 2, 3, 4, 5$-parton matrix elements are considered at NLO QCD. The merging parameter is set to $y_{\text{cut}} = (Q_{\text{cut}}/E_{\text{CMS}})^2 = 10^{-2.25}$. We evolve the strong coupling at the two-loop order, assuming $\alpha_s(m_Z) = 0.117$.[4] While for the cluster fragmentation model we have kept all relevant parameters at their default values, we have set the main parameters of the Lund model to

$$a = 0.3 \, (\texttt{PARJ(41)}), \quad b = 0.6 \, \text{GeV}^{-2} \, (\texttt{PARJ(42)}), \quad \sigma = 0.36 \, \text{GeV} \, (\texttt{PARJ(21)}). \tag{19}$$

For both hadronisation models a satisfactory agreement with data is observed. The variations between the two predictions stay within the few percent range.

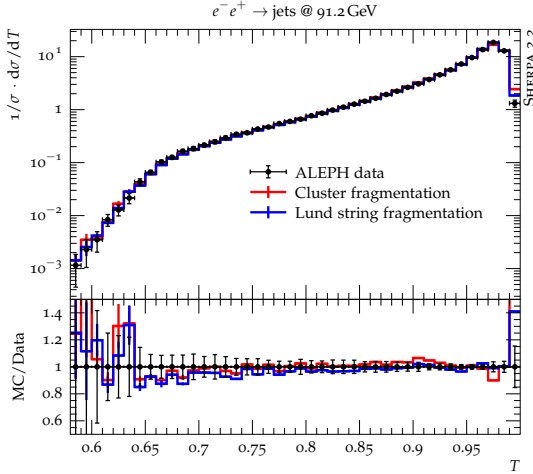

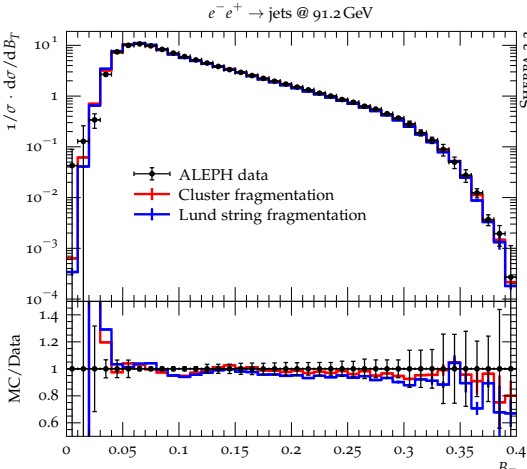

Figure 13: Results for the thrust distribution in jet production at LEP for the two fragmentation models available in SHERPA in comparison with ALEPH data [193].

Figure 14: Results for the distribution of the total jet broadening at LEP for the two fragmentation models available in SHERPA in comparison with ALEPH data [193].

**Underlying Event** As explained in Section 2.7, an accurate simulation of the effects of secondary scattering and their subsequent evolution is necessary to be able to describe observables at hadron colliders. As an example for an observable that is impacted by non-perturbative contributions from the underlying event and hadronisation we consider the differential jet-shape variable $\rho(r)$. In Fig. 15 we compare SHERPA particle-level predictions based on the two-jet-production matrix element, evolved by the CSSHOWER, and including effects from multiple parton scatterings and hadronisation. We give predictions based on three different PDF sets – consistently used throughout the hard process, initial-state parton showering and the underlying-event simulation – namely NNPDF 3.0 NNLO [159], MMHT 2014 NNLO [168] and CT14 NNLO [194]. The predictions for various slices of jet transverse momentum are

---

[4] Note, the SHERPA default value is $\alpha_s(m_Z) = 0.118$. However, we observed a marginally better description of LEP observables, and in particular the thrust distribution, both for the cluster and the Lund string fragmentation using $\alpha_s(m_Z) = 0.117$.

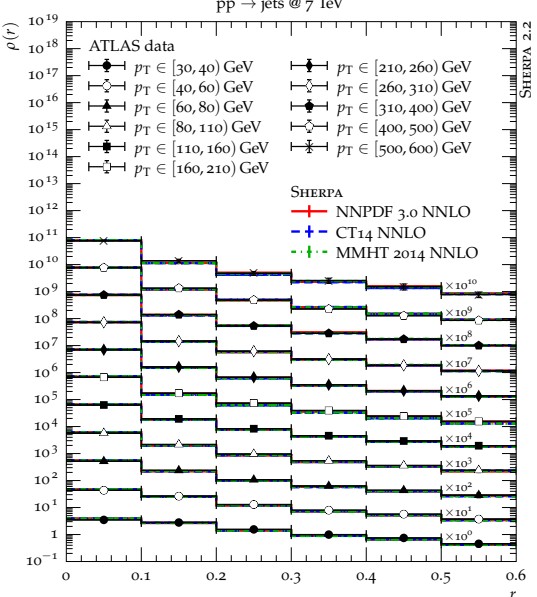

Figure 15: Results for the differential jet shape $\rho(r)$ in dependence on the jet transverse momentum within $|\eta| < 2.8$ in comparison to ATLAS data [195].

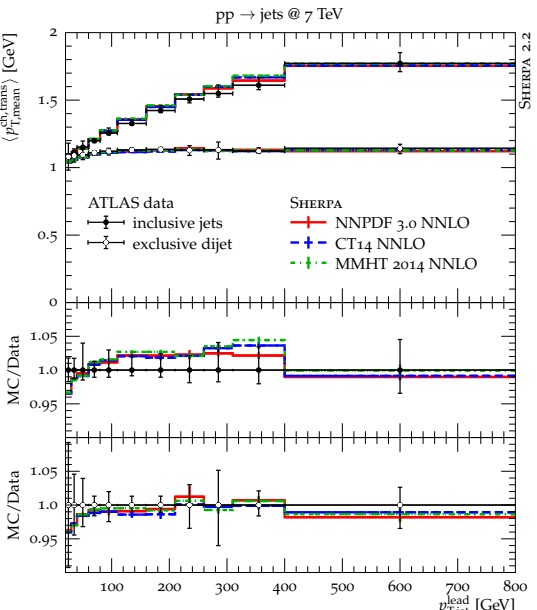

Figure 16: Results for the average mean charged-particle transverse momentum in the transverse region in dependence of the leading jet $p_T$ in inclusive jet and exclusive dijet events in comparison to data from the ATLAS experiment [196].

compared with data from the ATLAS collaboration taken in LHC Run 1 at $\sqrt{s} = 7$ TeV [195]. Notably, the predictions for all three PDF sets largely agree and yield a satisfactory description of the measurements. Please note that SHERPA's non-perturbative models have only been tuned using the NNPDF 3.0 NNLO set, and thus this level of agreement is non-trivial. Furthermore, jets at different transverse momenta receive different contributions from hadronisation and the underlying event. Clearly, the softer the jet, the larger the non-perturbative corrections the jet shape $\rho(r)$ receives.

As a second example we consider in Fig. 16 a more exclusive observable, namely the average of the mean charged-particle transverse momentum per event in the region transverse to the leading jet, differential in the leading-jet transverse momentum. This transverse region, defined with respect to the azimuthal angle of the leading jet as $60° < |\Delta\phi| < 120°$, is expected to be very sensitive to the underlying event. The fact that it measures charged particles only, renders it sensitive to the flavour structure of the hadronisation simultaneously. This observable has been measured by the ATLAS collaboration in [196], where results for inclusive jet and exclusive dijet production have been presented. Jets thereby have to fulfill $p_{T,j} > 20$ GeV and $|y_j| < 2.8$, for the charged particles in the transverse region it is required that $p_T > 0.5$ GeV and $|\eta| < 2.5$. The SHERPA predictions are in good agreement with the data, both for the inclusive jet and the exclusive dijet selection. No significant dependence on the PDF set employed in the simulation can be observed, despite SHERPA only having been tuned using one of the PDF sets, as discussed above.

**Hadron and $\tau$ Decays** As a last example, we show results which are sensitive to the modelling of hadron and $\tau$-lepton decays. The distribution of momentum transfer in the semileptonic $B^0 \rightarrow \pi^- e^+ \nu_e$ decay for various form-factor models implemented in SHERPA is com-

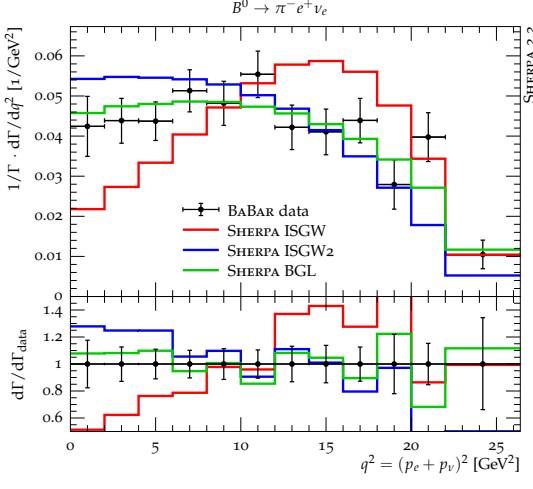
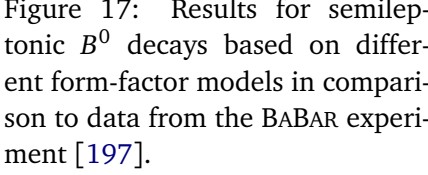
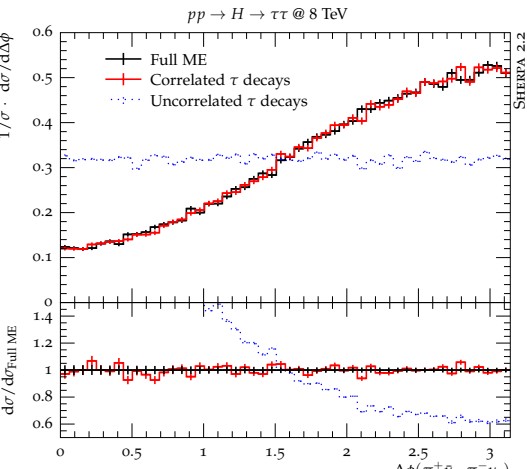

Figure 17: Results for semileptonic $B^0$ decays based on different form-factor models in comparison to data from the BABAR experiment [197].

Figure 18: Different predictions for the decay-plane angle between hadronically decaying $\tau$-leptons from Higgs-boson decays.

pared with data taken from the BABAR experiment [197] in Fig. 17. The BGL parametrisation [198] with parameters extracted from that measurement reproduces the data well, while the ISGW2 [199] and ISGW models [200] with their original parameter sets do not agree well with this measurement.

To illustrate the simulation of hadronic $\tau$ decays, we consider in Fig. 18 the production of Higgs bosons at the LHC which decay to a pair of $\tau$-leptons, which then are assumed to subsequently decay via $\tau \to \pi \nu$. Clearly, the effect of including spin correlations in the decay chain can have a dramatic effect when measuring correlations of the decay products, as exemplified for the decay-plane angle in the $H \to \tau(\to \pi \nu)\tau(\to \pi \nu)$ final state. In addition, one can see how the proper inclusion of spin-correlation effects leads to an excellent agreement with the full result, obtained with exact matrix elements for the decayed final state.

## 4 Conclusions

We have summarised essential features and improvements of the SHERPA 2.2 event generator. The SHERPA framework has been extensively used for event generation during the LHC Run 1 and Run 2, and represents a decade of developments towards ever higher precision in the simulation.

Key building blocks of the SHERPA generator are implementations of all the physics aspects needed for a full event description, including automated matrix-element generators, parton showers, a hadronisation model and a simulation of multiple parton interactions. Supplemented by methods to deal with particle decays, QED corrections and a large variety of interfaces, *e.g.* to parton density functions, New Physics models or event-output formats, this qualifies SHERPA as a full-fledged multi-purpose event generator for the modelling of scattering events at past, current, and future collider experiments. Certainly a highlight and unique feature of SHERPA are its comprehensive methods to combine higher-order perturbative matrix-element calculations with parton-shower simulations and especially its automation of the MEPS@NLO method. In Sec. 3 we have illustrated some applications of SHERPA to challenges posed in particular by the LHC experiments. Through the inclusion of exact NLO QCD

matrix elements the jet activity accompanying signal processes gets adequately modelled and at the same time theoretical uncertainties will be systematically reduced.

At this time development is moving towards SHERPA 3.0.0, heralding a major new development effort with exciting improvements across the board. They will include the description of higher-orders in the perturbative parts of the simulation, for example incorporating advances in the resummation properties of parton showers [20, 21, 119], the inclusion of approximate high-energy EW effects as based on [201], a fully massive five-flavour scheme for heavy quarks in the initial state [112], the improvements in efficient phase-space sampling [202], and extensions such as the automation of QCD soft-gluon resummation at NLL accuracy following the CAESAR formalism [203, 204]. This will be supplemented with continuous improvements in the non-perturbative modelling, such as an improved cluster hadronisation or a new model for inclusive QCD scattering.

With the preparations for LHC Run 3 in full swing, and many measurements with the full Run 2 data to appear in the next years, this new version will play an important role in the future analysis of LHC data.

## Acknowledgements

We would like to thank all our former collaborators on the SHERPA project, in particular the various students that contributed with their theses works to further improve our physics models and the corresponding computer codes. We are indebted to our users that have helped us to find and fix short-comings in the program.

This work has received funding from the European Union's Horizon 2020 research and innovation programme as part of the Marie Skłodowska-Curie Innovative Training Network MCnetITN3 (grant agreement no. 722104). It was supported by the U.S. Department of Energy under contract DE-AC02-76SF00515. It used resources of the Fermi National Accelerator Laboratory (Fermilab), a U.S. Department of Energy, Office of Science, HEP User Facility. Fermilab is managed by Fermi Research Alliance, LLC (FRA), acting under Contract No. DE–AC02–07CH11359.

SS acknowledges support through the Fulbright-Cottrell Award and from BMBF (contracts 05H15MGCAA and 05H18MGCA1). FS's research was supported by the German Research Foundation (DFG) under grant No. SI 2009/1-1. MS acknowledges the support of the Royal Society through the award of a University Research Fellowship. The work of DN is supported by the French Agence Nationale de la Recherche, under grant ANR-15-CE31-0016.

## A    Default-tune non-perturbative model parameters

The major parameters specifying the cluster-fragmentation model in SHERPA and their default values obtained by tuning in particular to LEPI data:

```
(fragmentation){
% strange quark fraction
STRANGE_FRACTION 0.6049;
% baryon fraction
BARYON_FRACTION 1.0;
% various quark, diquark flavour selection weights
P_{QS}/P_{QQ} 0.3;
P_{SS}/P_{QQ} 0.01;
P_{QQ_1}/P_{QQ_0} 1.0;
```

```
% gluon splitting kinematics, \eta and p^2_{T,0}, cf. Eq. (2.4)
G2QQ_EXPONENT 1.08;
PT^2_0 1.56;
% cluster splitting exponents, cf. Eq. (2.10)
SPLIT_EXPONENT 0.1608;
SPLIT_LEAD_EXPONENT 1.0;
SPECT_EXPONENT 1.739;
SPECT_LEAD_EXPONENT 8.0;
% cluster to hadron decays
DECAY_OFFSET 1.202; % \kappa
DECAY_EXPONENT 2.132; % \chi
% multiplet weights
MULTI_WEIGHT_L0R0_VECTORS 0.75; % $\rho$, $K^*$, $\omega$, $\phi$ etc
MULTI_WEIGHT_L0R0_TENSORS2 0.30; % $a_2(1320)$, $f_2(1270)$, $f'_2(1525)$,
    $K^*_2(1430)$ etc
MULTI_WEIGHT_L0R0_DELTA_3/2 0.45; % $\Delta(1232)$, $\Sigma(1385)$,
    $\Xi(1530)$, $\Omega^-$ etc
% flavour-specific enhancement factors
HEAVY_CHARMBARYON_ENHANCEMENT 0.9;
HEAVY_BEUATYBARYON_ENHANCEMENT 1.7;
HEAVY_CHARMSTRANGE_ENHANCEMENT 1.0;
HEAVY_BEAUTYSTRANGE_ENHANCEMENT 3.0;
}(fragmentation)
```

Parameters for the intrinsic transverse-momentum distribution of initial-state partons in composite beam particles, cf. Eq. (3). Per default initial-state protons are assumed. In the case of proton–proton collisions the same values for the mean and width of the corresponding Gaussian distributions are assumed, respectively, *i.e.*

```
(beam){
% Gaussian distributed intrinsic k_T in initial-state protons
K_PERP_MEAN_1 1.1; % <k_T> beam 1
K_PERP_MEAN_2 1.1; % <k_T> beam 2
K_PERP_SIGMA_1 0.85; % \sigma beam 1
K_PERP_SIGMA_2 0.85; % \sigma beam 2
}(beam)
```

These values correspond to the reference collision energy of $E_{\mathrm{ref}} = 7$ TeV. For the width parameter the value at other centre-of-mass energies is computed as

$$\sigma(E_{\mathrm{cms}}) = \sigma(E_{\mathrm{ref}})\left(\frac{E_{\mathrm{cms}}}{E_{\mathrm{ref}}}\right)^{0.55}. \tag{20}$$

List of major parameters defining the multiple-parton interaction model in SHERPA and their default-tune values, cf. Eq. (1):

```
(mi){
% MPI model parameters
SIGMA_ND_FACTOR 0.3142; % factor multiplying non-diffractive cross section
% cut-off parameter and regulator, given at reference scale E_{ref} = 1.8
    TeV
SCALE_MIN 2.895; % p_{T,min}(E_{ref})
TURNOFF 0.7549; % p_{T,0}(E_{ref})
TURNOFF_EXPONENT 0.244; % \alpha
}(mi)
```

The parameters $p_{T,\min}(E_{\mathrm{ref}})$ and $p_{T,0}(E_{\mathrm{ref}})$ are evolved from the reference energy $E_{\mathrm{ref}}$ (REFERENCE_SCALE) to the actual collider energy $E$ according to

$$p_{T,i}(E) = p_{T,i}(E_{\mathrm{ref}})\left(\frac{E}{E_{\mathrm{ref}}}\right)^{\alpha} , \tag{21}$$

with the power $\alpha$ specified by the parameter TURNOFF_EXPONENT.

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
