# Peer review of "Event Generation with Sherpa 2.2"

_SciPost Physics, doi:SciPost Phys. 7, 034 (2019)_

## Round 1 · Referee Report · Anonymous (Referee 1) · 2019-6-28

Report

The paper serves as a summary of the Sherpa general-purpose MC event generator as of version 2.2.6. It presents a discussion of the various components of the modular generator, including pseudocode example for their invocation, along with a number of physics results achieved with Sherpa relevant for the current experimental program at the LHC.
The wide use by the community of Sherpa both presently and in the foreseeable future makes the existence of a canonical publication giving to cite when quoting results obtained from its latest versions important in ensuring that Sherpa development gets its proper due, as well as providing an efficient summary of its abilities. The presentation is clear and well-formatted throughout. I will have no problem recommending publication, as soon as the following requests for clarification and editing corrections, all of which are of a minor nature and should be straightforward to deal with, are addressed.

Requested changes

Clarification of physics points: 1) Sec. 2.2, CSShower. Finite-mass effects are brought up in the context of b-quark studies in the 4-/5-flavour schemes. Is there a technical obstruction to implementing variable-flavour schemes or should they be mentioned as well? 2) Sec. 3.3. It would it interesting to know if there are any known distributions in which differences between the SM and HEFT predictions that due depend significantly on the number of jets, especially as this is one of the few result sections that does not have a corresponding extended study that has been published elsewhere. 3) Sec. 3.7. While MEPS@NLO does give good overall agreement to the data, both MEPS@(N)LO seem to disagree with the data by the same feature near EγT70 GeV. Is the origin of this structure known? 4) Sec. 3.10. For the statement comparing BGL and ISGW(II) parameterizations in semileptonic B decays, can we authors clarify where the choice of model parameters in the comparison originates from, i.e., are the form factors used in the comparison extracted in a consistent manner from the same data?

Minor grammatical corrections: 1) Throughout the paper, the construction "...allow to [VERB].." is used, e.g. "Sampling ... allows to determine...". This is not a grammatical English construction and should be replaced by something like "...allow [NOUN] to [VERB]..." (...allows us/the user to determine...) or a verbal noun in place of the verb (...allows for the determination of...). 2) Sec. 2.1, Built-in MEGs: "Both MEGs are capable to simulate..." "Both MEGs are capable of simulating..." 3) Sec. 2.1, NNLO QCD Calulations: "...can be in the form of plugins obtained..." "...can be obtained in the form of plugins..."

  • validity: high
  • significance: high
  • originality: good
  • clarity: high
  • formatting: excellent
  • grammar: good

Author:  Steffen Schumann  on 2019-09-05  [id 594]

(in reply to Report 1 on 2019-06-28)

We would like the referee for the careful and detailed consideration of our manuscript. Addressing the comments and requests helped us improve the quality of the paper. Accordingly, we have submitted a revised version to the arXiv. The detailed replies and the corresponding changes to the draft are listed below.

We hope that the new version meets the requirements and the paper can be published in its current form.

[referee] Clarification of physics points: 1) Sec. 2.2, CSShower. Finite-mass effects are brought up in the context of b-quark studies in the 4-/5-flavour schemes. Is there a technical obstruction to implementing variable-flavour schemes or should they be mentioned as well?

[reply] The currently implemented shower scheme is, in fact, a general-mass variable flavour-number scheme (VFNS) complemented with a fully massive phase space implementing, among other effects, the massive quark thresholds. A sentence to this effect has been added to the manuscript.

[referee] 2) Sec. 3.3. It would it interesting to know if there are any known distributions in which differences between the SM and HEFT predictions that due depend significantly on the number of jets, especially as this is one of the few result sections that does not have a corresponding extended study that has been published elsewhere.

[reply] In further studies, e.g. reference [83] and yet unpublished material, we see very few appreciably jet-multiplicity dependent mass effects. We have added two sentences detailing this to the Higgs section.

[referee] 3) Sec. 3.7. While MEPS@NLO does give good overall agreement to the data, both MEPS@(N)LO seem to disagree with the data by the same feature near ETγ ∼70 GeV. Is the origin of this structure known?

[reply] We are not aware of any particular physics reason for this structure. But it is worth noting, that the systematic uncertainties of measurement and MEPS@NLO prediction overlap.

[referee] 4) Sec. 3.10. For the statement comparing BGL and ISGW(II) parameterizations in semileptonic B decays, can we authors clarify where the choice of model parameters in the comparison originates from, i.e., are the form factors used in the comparison extracted in a consistent manner from the same data?

[reply] They are indeed not extracted from the same data. For the BGL parametrisation more recent data [189] was included in the extraction. We have clarified this in the text.

[referee] Minor grammatical corrections: ...

[reply] We have implemented all the suggested corrections.

Author:  Steffen Schumann  on 2019-07-18  [id 568]

(in reply to Report 1 on 2019-06-28)
Category:
remark

We would like to thank the referee for his/her comments on our manuscript. We will address all points raised in a more detailed reply and an revised version of the paper. Before doing that we would like to await the other referee's comments.

All the best,
Steffen Schumann (for the authors)

---

## Round 1 · Referee Report · Anonymous (Referee 2) · 2019-7-24

Report

The manuscript describes the main features of the event generator package Sherpa as of version 2.2. As such it documents the progress made in developing the Sherpa event generator and references the main physics publications which document the various features in more detail. It is well structured and presented.

Such manuals or reviews provide important reward for the many man-years of work which went into the development of multi-purpose event generators and as such deserve proper publication. I am happy to recommend the manuscript for publication in SciPost once a few formulations and minor details are addressed.

In section 2, most of the subsections provide enough references for the reader to find more details if needed, however some sections do not and thus require somewhat more details to be highlighted, also addressing the specific implementation in Sherpa. This applies to the section on hadronization, and on multi-parton interactions and beam remnants, where the latter miss explicit formulae e.g. of what distribution the intrinsic kt is drawn from.

Regarding the CSShower algorithm more details are required concerning the evolution of massive quarks, and other work in this context should be referenced, as well. The description of multi-jet merging needs to discuss how the Sherpa approach compares to algorithms implemented in other generators.

In section 3 more details regarding the choice of strong coupling need to be added, and specifically how this relates to tuning in the presence or absence of muti-jet merged calculations to shed more light on the choice made. At least a comparison line or a change in goodness-of-fit needs to be mentioned to quantify the impact. More details on the tuning procedure are required, as well.

It is clear that a document like the one at hand mainly serves the purpose of highlighting the features of the program and future
strategies, however this should purely happen through the presentation of the scientific content and I felt that some formulations where slightly too exaggerated and not appropriate. Examples are in the introduction ("high-precision predictions" today refers to different calculations than the ones delivered even by highly advanced event generators), more examples are in the third paragraph of the conclusion, and in between the availability of "a number of" NNLO calculations are mentioned, which is simply not the right wording for a total of three processes.
  • validity: -
  • significance: -
  • originality: -
  • clarity: -
  • formatting: -
  • grammar: -

Author:  Steffen Schumann  on 2019-09-05  [id 593]

(in reply to Report 2 on 2019-07-24)
Category:
answer to question
correction

We would like the referee for the careful and detailed consideration of our manuscript. Addressing the comments and requests helped us improve the quality of the paper. Accordingly, we have submitted a revised version to the arXiv. The detailed replies and the corresponding changes to the draft are listed below.

We hope that the new version meets the requirements and the paper can be published in its current form.

[referee] In section 2, most of the subsections provide enough references for the reader to find more details if needed, however some sections do not and thus require somewhat more details to be highlighted, also addressing the specific implementation in Sherpa. This applies to the section on hadronization, and on multi-parton interactions and beam remnants, where the latter miss explicit formulae e.g. of what distribution the intrinsic kt is drawn from.

[reply] We have significantly extended the presentation of the non-perturbative models in Sec. 2. We furthermore added a section that give some detail on the tuning of model parameters and provide a list of the default-tune setting in new appendix.

[referee] Regarding the CSShower algorithm more details are required concerning the evolution of massive quarks, and other work in this context should be referenced, as well. The description of multi-jet merging needs to discuss how the Sherpa approach compares to algorithms implemented in other generators.

[reply] We have added a comment on the treatment of quark masses the the text, see above. However, given the purpose of the paper on the presentation of the status of matching and merging algorithms available in Sherpa, we do not feel that an extended review and comparison to other solutions on the market would be adequate here. This information should be well covered by the references provided detailing and comparing the corresponding methods developed by the Sherpa authors.

[referee] In section 3 more details regarding the choice of strong coupling need to be added,

[reply] For the LEP setup, the information is already contained in Sec. 3.10. For all other setups (which are using PDFs), we have added a paragraph in Sec. 2.5 explaining that by default the choice of the strong coupling is given by the PDF set. Furthermore, in the new Sec. 2.10 this aspect is also stressed.

[referee] and specifically how this relates to tuning in the presence or absence of muti-jet merged calculations to shed more light on the choice made. At least a comparison line or a change in goodness-of-fit needs to be mentioned to quantify the impact. More details on the tuning procedure are required, as well.

[reply] As already mentioned, we have introduced an entire new section on tuning, collating the default-tune parameters in the new App. A.

[referee] It is clear that a document like the one at hand mainly serves the purpose of highlighting the features of the program and future strategies, however this should purely happen through the> presentation of the scientific content and I felt that some formulations where slightly too exaggerated and not appropriate. Examples are in the introduction ("high-precision predictions"> today refers to different calculations than the ones delivered even by highly advanced event generators), more examples are in the third paragraph of the conclusion, and in between the av> ailability of "a number of" NNLO calculations are mentioned, which is simply not the right wording for a total of three processes.

[reply] We have toned down our statements correspondingly.

---

## Round 2 · Author Response

We would like the referees for their careful and detailed considerations of our manuscript. Addressing the comments and requests helped us improve the quality of the paper. Accordingly, we have submitted a revised version to the arXiv. The detailed replies and the corresponding changes to the draft or listed below.
We hope that the new version meets the requirements and the paper can be published in its current form.
We hope that the new version meets the requirements and the paper can be published in its current form.

---

## Round 2 · List of Changes

In addressing the referees comments we have extended the presentation of the non perturbative model in Sherpa, in particular Sec. 2.7 and 2.8. Furthermore, we added a section on tuning of the various model parameters and now provide an appendix that lists the default-tune parameters.
We have furthermore clarified the treatment of mass effects in the parton shower. Also, we more clearly state the default assumed for the strong coupling in the simulations.
Lastly, we have fixed a number of minor grammatical mistakes.
We have furthermore clarified the treatment of mass effects in the parton shower. Also, we more clearly state the default assumed for the strong coupling in the simulations.
Lastly, we have fixed a number of minor grammatical mistakes.

---

## Editorial Decision

published